# A Review of Deep Learning-Based LiDAR and Camera Extrinsic Calibration

**DOI:** 10.3390/s24123878

**Published:** 2024-06-15

**Authors:** Zhiguo Tan, Xing Zhang, Shuhua Teng, Ling Wang, Feng Gao

**Affiliations:** 1School of Electronic Information, Hunan First Normal University, Changsha 410205, China; zhangxing21@hnfnu.edu.cn (X.Z.); wangling@hnfnu.edu.cn (L.W.); gaofeng_nudt@hnfnu.edu.cn (F.G.); 2Key Laboratory of Hunan Province for 3D Scene Visualization and Intelligence Education, Changsha 410205, China; 3Zhongwei Xinzhi (Chengdu) Technology Co., Ltd, Chengdu 610213, China

**Keywords:** extrinsic parameter calibration, deep learning, LiDAR, camera, data fusion

## Abstract

Extrinsic parameter calibration is the foundation and prerequisite for LiDAR and camera data fusion of the autonomous system. This technology is widely used in fields such as autonomous driving, mobile robots, intelligent surveillance, and visual measurement. The learning-based method is one of the targetless calibrating methods in LiDAR and camera calibration. Due to its advantages of fast speed, high accuracy, and robustness under complex conditions, it has gradually been applied in practice from a simple theoretical model in just a few years, becoming an indispensable and important method. This paper systematically summarizes the research and development of this type of method in recent years. According to the principle of calibration parameter estimation, learning-based calibration algorithms are divided into two categories: accurate calibrating estimation and relative calibrating prediction. The evolution routes and algorithm frameworks of these two types of algorithms are elaborated, and the methods used in the algorithms’ steps are summarized. The algorithm mechanism, advantages, limitations, and applicable scenarios are discussed. Finally, we make a summary, pointing out existing research issues and trends for future development.

## 1. Introduction

With the gradual maturity and commercialization of LiDAR technology, LiDAR and camera have become the most common and fundamental sensor combination in unmanned driving and autonomous robot navigation, and are currently the most typical multi-sensor fusion devices. This technology has been widely used in various challenging tasks, such as object detection and tracking [1,2,3,4,5,6,7,8], simultaneous localization and mapping [9,10,11], and autonomous navigation [12,13]. In order to achieve effective and accurate data fusion, it is necessary to calibrate the extrinsic parameters of LiDAR and the camera, namely to estimate the transformation between the two sensors in a common reference frame. It is a fundamental issue in fields such as computer vision, autonomous robot navigation, optical measurement, and autonomous driving. Especially in the application research of autonomous robot navigation and autonomous driving, many scholars have achieved remarkable results in recent decades and have also achieved successful applications [14,15].

Since data from different sensors vary in, e.g., size, resolution, field of view, etc., they are a great challenge for extrinsic parameter calibration without a clear target, especially in the actual environment. In recent years, with successful application in various fields, deep learning has also been applied in LiDAR and camera calibration, demonstrating powerful abilities and becoming one of the important methods. In this paper, we summarize and review applications and developments of deep learning-based LiDAR and camera calibration methods in recent years.

Although there have been some reviews on LiDAR and camera calibration, such as Nie et al. [1], Wang et al. [2], Li et al. [4], Liu et al. [16], Khurana et al. [17], which all contain content about learning-based methods, they only provide a relatively brief summary. With the rapid development of this technology, we believe that it is necessary and important to conduct a comprehensive and in-depth review on this topic. The contributions of this paper are as follows:(1)As we know, this is the first independent review on deep learning-based extrinsic calibration, which can provide significant assistance and reference for researchers engaged in this field.(2)We provide a novel classification of LiDAR–camera extrinsic calibration based on deep learning. Different from the previous works, we divide the methods into two categories, namely, accurate extrinsic parameters estimation (AEPE) and relative extrinsic parameters prediction (REPP).(3)We provide a comprehensive and detailed explanation of the existing works. We not only study the different types of methods holistically, but also provide the classification and summary of the procedures in the algorithms step by step.(4)In previous research, they usually summarized deep learning calibration algorithms as black boxes, one by one. Different from those works, we have conducted a rational phased and modular study and summary on deep learning-based calibration algorithms, which is more conducive for researchers, especially those who are new to related research, to understand calibration and its deep learning methods from a theoretical perspective.(5)We also provide a detailed comparison and discussion on different methods, including their evolution process, principles, characteristics, advantages, and limitations.

## 2. Problem Formulation

The LiDAR and camera calibration is to obtain the extrinsic parameters between the sensors by processing the point clouds and RGB images from them to obtain the transformation, thereby transforming the data into the same coordinate system. The calibration is crucial for improving the performance of perception tasks, such as object detection, classification, tracking, etc. [18,19,20].

In this section, we review the transformation relationship between a LiDAR coordinate system and a camera coordinate system. Broadly speaking, the calibration of two sensors includes both temporal and spatial calibration. This paper is focused on the calibration in space, which is to obtain the rigid transformation between the two sensors. We also assume that temporal calibration has been performed well just like other papers [1,4,16].

### 2.1. Transformation between LiDAR and Camera Coordinates

The 3D points set $P={\left[P_{1},P_{2},\cdots ,P_{n}\right]}^{T}$, is denoted as $P^{C}={\left[P_{1}^{C},P_{2}^{C},\cdots ,P_{n}^{C}\right]}^{T}$ in the camera coordinate system and $P^{L}={\left[P_{1}^{L},P_{2}^{L},\cdots ,P_{n}^{L}\right]}^{T}$ in the LiDAR coordinate system. According to the classical pinhole camera model, any point $P_{i}^{C}=\left(X_{i}^{C},Y_{i}^{C},Z_{i}^{C}\right)$ is projected onto the camera image plane as a 2D point $p_{i}^{C}=\left(x_{i}^{C},y_{i}^{C}\right)$:(1)$$Z_{i}^{C}\left[\begin{array}{c} x_{i}^{C}\\ y_{i}^{C}\\ 1 \end{array}\right]=\left[\begin{array}{ccc} f_{x} & s & x_{0}\\ 0 & f_{y} & y_{0}\\ 0 & 0 & 1 \end{array}\right]\left[\begin{array}{c} X_{i}^{C}\\ Y_{i}^{C}\\ Z_{i}^{C} \end{array}\right]=K\left[\begin{array}{c} X_{i}^{C}\\ Y_{i}^{C}\\ Z_{i}^{C} \end{array}\right],$$
where K is the camera intrinsic parameters matrix, $f_{x},f_{y}$ are the focal lengths in pixels on the x and y axes, respectively, $\left(x_{0},y_{0}\right)$ is the principal point, and s is the skew coefficient. Let $T_{\phi }=\left[\begin{array}{cc} R & t \end{array}\right]$ be the transformation between LiDAR and camera, which is determined by the extrinsic parameters ϕ={R,t}. The corresponding points $P_{i}^{L}=\left(X_{i}^{L},Y_{i}^{L},Z_{i}^{L}\right)$ in the LiDAR coordinate system can be transformed to the camera coordinate system as follows:(2)$$\left[\begin{array}{c} X_{i}^{C}\\ Y_{i}^{C}\\ Z_{i}^{C} \end{array}\right]=R\left[\begin{array}{c} X_{i}^{L}\\ Y_{i}^{L}\\ Z_{i}^{L} \end{array}\right]+t=\left[\begin{array}{cc} R & t \end{array}\right]\left[\begin{array}{c} X_{i}^{L}\\ Y_{i}^{L}\\ Z_{i}^{L}\\ 1 \end{array}\right]$$

From Equations (1) and (2), we can obtain the transformation relationship between the LiDAR points and the 2D pixels in the camera images. Based on this transformation relationship, the LiDAR points can be projected onto the camera imaging plane to obtain the corresponding LiDAR depth map D and the pixels, $p_{i}^{L,D}$ abbreviated as $p_{i}^{L}=\left(x_{i}^{L},y_{i}^{L}\right)$.
(3)$$Z_{i}^{C}\left[\begin{array}{c} x_{i}^{L}\\ y_{i}^{L}\\ 1 \end{array}\right]=KT_{\phi }\left[\begin{array}{c} X_{i}^{L}\\ Y_{i}^{L}\\ Z_{i}^{L}\\ 1 \end{array}\right]=P_{roject}\left(\phi ,K,P_{i}^{L}\right)$$

It is worth noting that some LiDAR not only obtain range but also intensity. By using the above method, we can also obtain the intensity map N, and its pixel $p_{i}^{L,N}$. The transformation between the coordinate systems and data are shown in Figure 1.

### 2.2. Classification of Calibration Methods

According to the need for calibration targets, extrinsic calibration between a LiDAR sensor and a camera can be divided into the following two categories [1,2,4]:

**Target-based calibration.** By using artificial objects as multi-sensor co-shooting targets, the corresponding relationship between point clouds and RGB images is constructed to obtain extrinsic parameters. There are various objects used as calibration targets, such as checkerboards [21,22], ARUCO (Augmented Reality University of Cordoba) tags [23,24], special calibration boards [25,26,27], reflective labels [28], laser calibration cards [29], and so on.

**Targetless calibration.** Targetless calibration methods estimate extrinsic parameters by exploiting useful information from surroundings environments automatically. It has been widely applied in multi-sensor systems such as autonomous driving [20,30,31], and is currently a challenging hotspot in academic research. Targetless calibration was divided into four classes in paper [4], namely, information theory-based, feature-based, ego-motion-based, and learning-based methods. This classification is a non-mutually exclusive classification method, which has intersections between methods, as shown in Figure 2. For example, SemCal [32] is a semantic feature-based method, but extracts features by learning networks and uses information theory-based methods to estimate extrinsic parameters. For another example, ego-motion-based methods in papers [33,34], use information theory-based and feature-based methods for the accurate estimation of extrinsic parameters, respectively. This paper summarizes the extrinsic calibration methods based on deep learning. The deep learning-based method first appeared in 2017. In just a few years, this kind of method has become one of the important means in calibration, with great potential. In the existing review literature [1,2,4,17], the calibration methods have been comprehensively summarized, and all introduced such types of methods at the same time. Based on the architecture of the method, Li et al. [4] classified deep learning-based methods into two categories: end-to-end methods and hybrid learning methods. Although this classification is simple and clear, it is not conducive to readers understanding the original ideas from the perspective of the underlying principles. Wang et al. [2] classified them as semantic feature-based methods and scenario-independent methods; Luo et al. [1] classified them as feature-based and other methods. 

Different from previous works, we only focus on deep learning-based extrinsic calibration methods and conduct more reasonable classification. We deeply analyze the characteristics of each category and consider its potential applications in various autonomous systems.

## 3. Deep Learning-Based Extrinsic Calibration

From existing public literatures, we divide learning-based calibration methods into two categories based on the principle of extrinsic parameters calculation: AEPE (Accurate Extrinsic Parameters Estimation), and REPP (Relative Extrinsic Parameters Prediction), as shown in Figure 3.

The AEPE methods are currently mainstream methods. An AEPE method establishes correspondences between features in the current observational data of the LiDAR and camera, and then performs extrinsic parameters estimation. According to features used in the algorithm, AEPE methods can also be divided into global feature-based methods [35] and local feature-based methods [36]. Different from the AEPE, an REPP method retrieves a historical calibrated gallery set using current observational data to obtain relevant candidate data, and predicts the extrinsic parameters by fusing the retrieved data [37]. Currently, there is relatively little research on this type of method and it is still in its infancy.

Therefore, this section will be expanded according to the following structure, as shown in Figure 3. Due to more scholars diving into research on the AEPE, a relatively mature framework has formed. These methods will be studied in Section 3.1. Section 3.2 and Section 3.3 will, respectively, discuss the AEPE methods based on global features and local features. In Section 3.4, we expound on the REPP and elaborate on its origin, basic principles, and specific implementation.

### 3.1. Accurate Extrinsic Parameters Estimation

The general process of an AEPE method can be described as: First, LiDAR and camera acquire two data branches, i.e., point clouds and RGB images. Second, the AEPE finds the matches between two branches. Lastly, the AEPE estimates the final external parameters according to the matches under the framework of the calibration network or the calibrating optimization model.

Typical methods include several tasks such as feature extraction, feature matching, parameter regression/optimization, and loss assessment [35,38]. Deep learning networks complete one or several tasks within them. Generally speaking, in order to maintain the consistency of extracted features from different data branches, a data conversion (data preparation [39,40,41,42,43]) is usually performed before data input. The basic process of an AEPE method is shown in Figure 4. According to the features, AEPE methods are divided into two categories: global feature-based and local feature-based methods.

**Global feature-based method**. Global feature refers to the feature extracted by convolution neural network (CNN)-based networks, which convolute and process the entire image to obtain a global representation of the image [35,44,45,46,47,48]. This global feature is different from the traditional ones, such as color (or intensity), texture, and shape, and is an abstract feature which is the overall attribute transformed from the traditional global feature by the network. A global feature-based method is the one calibrating extrinsic parameters between LiDAR and camera using these attributes.

**Local feature-based method.** Local features include high-level features and low-level features. High-level features refer to features close to human understanding, such as car [49], pedestrian [36], road [50], etc., in the image and the point cloud [36,51,52]. Low-level features refer to features such as contour [53], edge [54], line [55], and key point [56]. A calibration method based on local features is a method of using these kinds of features for extrinsic calibration [36].

As shown in Figure 4, there are two kinds of architecture, namely the end-to-end network and the hybrid learning method.

In end-to-end mode [44,57], deep learning networks complete the basic tasks from feature extraction to loss assessment in AEPE, such as CalibNet [44], which completes the entire process of accurate calibration and estimation except for the data conversion step (which some also include). Its backbone network mainly completes feature extraction, feature matching, and global regression tasks. The end-to-end mode is widely used in global feature-based calibration.

In hybrid learning mode, the calibration is usually divided into several stages [32,36,50,58], and the backbone network only completes feature extraction [36], as shown in the dashed box labeled one in Figure 4, or feature extraction and matching tasks, such as CFNet [59] and DXQ-Net [41], as shown in a dashed box labeled two in Figure 4. Feature matching may not be a necessary step throughout the calibration process, but may appear implicitly or be integrated into the optimization. For example, in SOIC [36], the backbone network is used to extract semantic features, while the matching process utilizes semantic attributes directly for implicit correspondence. In SemCal [32], the matching process is optimized together with parameters estimation in a mutual information model. This hybrid learning mode is commonly used in local feature-based methods.

### 3.2. Global Feature-Based Method

A typical global feature-based method uses a CNN-based encoder and decoder module to extract global feature maps from depth maps and RGB images, performs global feature matching using a convolution or correlation layer, and finally regresses extrinsic parameters. By using a CNN network to extract global features in this type of method, there are few requirements for scene structure, but its noise resistance and adaptability to environmental changes need to be improved.

RegNet [35] was the first deep learning network proposed by Nick Schneider et al. in 2017 for LiDAR and camera extrinsic calibration, but they did not consider spatial geometric relationships and only used quaternions as loss function. In response to this issue, Iyer et al. proposed CalibNet [44] in 2018, which designed loss functions using photometric and point cloud distance. Taking into account geometric transformation factors, they improved the calibration accuracy, but the network is sensitive to noise. On the basis of the above research, Yuan et al. [45] argued that the Euclidean geometric distance description of calibration parameter loss is inaccurate, and proposed RGGNet, which uses Riemannian geometric theory to construct a loss function, but this does not fundamentally change the algorithm’s robustness. To enhance the accuracy and robustness of the algorithm, Zhao et al. used the ResNet structure for feature matching in CalibDNN [43] in 2021.

In 2021, Lv et al. proposed CFNet [59], and they believed that deep learning feature extraction was limited by single resolution feature information, and the combination of multi-resolution features could be more conducive to calibration. Therefore, referring to the optical flow estimation model, PWC-Net [60], the network extracted feature maps of different resolutions during the feature extraction stage, and matched the maps through multi-resolution correlation during the matching stage. Similarly, in 2022, Y. Wu et al. proposed the PSNet [61] model, in which a parallel combination structure was used.

Attention-based methods have also been introduced to improve the performance of calibration networks. In 2022, An et al. proposed LIA-SC-Net [38], in which they used LiDAR intensity data as a basis to introduce the attention mechanism and combined it with depth map features to obtain observable features in RGB images and improve the correct rate of matching. Since the attention mechanism can improve accuracy and efficiency, in 2022, Shang et al. proposed the CALNet [39] model, which integrates channel attention, a hybrid spatial pyramid pooling (HSPP), and a liquid time constant network (LTC) [62] into a unified framework. This not only improves the calibration accuracy of the framework, but also ensures the robustness of the model.

Considering that the calibration, especially online calibration, is a time-varying process with correlations between data frames, Shi et al. proposed the CalibRCNN [57] in 2020, applying a recurrent neural network, long short term memory network (LSTM), to extract temporal characteristics, and estimating extrinsic parameters through the constraint of the time. In 2022, Nguyen et al. replaced LSTM in CalibRCNN with a Bi-LSTM module and proposed CalibBD [63]. Similarly, Zhu et al. applied LSTM to point cloud depth maps and RGB depth maps in CalibDepth [47], thereby improving the accuracy and robustness of the algorithm. In 2022, Shang et al. also introduced the LTC network [62] into the calibration. In 2023, Liu et al. [64] proposed another straightforward approach, causal calibration. They took several consecutive frames of the corresponding depth map and RGB image pairs as the input, and optimized the result by iterative calibration. In the algorithm, the predicted parameters of the previous frame are used to pre-calibrate the input depth map of the next frame.

Different from the strategies mentioned above, Wang Shuo et al. proposed an algorithm for CF (calibration flow) estimation in 2021 in CFNet [59]. They defined the changes of the same corresponding pairs in depth maps and RGB images caused by calibration as CF, which is similar to the optical flow. Then, the problem was transformed to estimating the CF and obtaining the corresponding relationships between point clouds and RGB images. After that, the extrinsic parameters were estimated by the random sample consensus (RANSAC) method combined with the EPnP method [59]. Although the method provides a new theoretical concept, the essential issue of calibration remains unchanged. Similarly, in 2022, Jing et al. proposed DXQ-Net [41] with the same basic idea, but used a more advanced optical flow network, RAFT [65], to estimate CF.

In addition to single cameras, researchers also pay attention to the issue of multi-cameras. In 2021, Wu et al. proposed NetCalib [66] for the extrinsic calibration of binocular cameras and LiDAR, which used binocular-generated depth maps and LiDAR depth maps as input. Slightly different from previous models, they added a spatial pyramid pooling (SPP) layer [67] in the feature matching module to generate fixed size feature maps and to achieve the input with any size. The Calibration-Net [42] constructed by Duy et al. applied the LCCNet [46] model to calibrate with two depth map branches. And Zhang et al. [68] used RegNet as the backbone network, but improved it with geometric and photometric losses. The above methods convert RGB images into depth maps to make the camera and LiDAR data structure consistent, which enhance the performance of the calibration network. Similarly, in order to make data structures consistent, Cocheteux et al. adopted another approach in PseudoCal [69], which estimated the depth map from the RGB image and transformed it into point cloud, then two point clouds were inputted into the network for processing. The advantage of these methods is that the networks process the data with the same structure, which is beneficial to the stage of feature extraction and matching. However, the disadvantage lies in complex data preparation and, hence, lower efficiency. Moreover, during the conversion process, there are noise, voids, holes, and so on, which will affect the accuracy. 

In addition, most previous studies are not suited to low resolution LiDAR, which has appeared in many practical applications. To this end, Zhang et al. proposed an algorithm in 2023 to calibrate low resolution LiDAR and cameras [70]. In their method, depth interpolation is applied to increase point density, and supervised contrastive learning is used to enhance noise resistance.

#### 3.2.1. Basic Framework

A global feature-based method usually adopts an end-to-end network model. Although the data procedure of the network is considered as a “black box”, it can still be clearly divided into steps such as feature extraction, feature matching, parameters regression, and loss assessment. In order to obtain a consistent global feature representation, this type of method usually performs 3D–2D data conversion when the LiDAR point cloud data enter the model. In some studies, in order to better match the features extracted from the two types of data, RGB images are used to generate depth maps for feature extraction, as shown in Figure 5. 

**Data conversion.** In order to ensure consistency in subsequent feature extraction and accuracy in matching, data conversion is usually performed on point clouds by projecting them onto the image plane to obtain depth maps, which achieves data consistency in dimensions. In some algorithms, RGB images are also converted to depth maps or point clouds to meet the input requirements of designed networks.

**Feature extraction.** In this type of algorithm, two data streams typically perform global feature extraction on the input data through CNN-based networks with the same structure, which can obtain the global features (feature maps).

**Feature matching.** Through the matching network module, the correspondence relationship between feature maps is established by the correlation layer or convolution layer. The construction of this correspondence relationship is different from the traditional one, which generally does not provide one-on-one matching results, but provides feature maps aggregated by two data flows.

**Parameters regression.** Extrinsic parameters are predicted by a global regression module. In a typical end-to-end calibration network, a multilayer perceptron (MLP) is often used for regression. In a hybrid learning mode, a traditional optimization is generally used to solve the problem.

**Loss assessment.** In an end-to-end network mode, loss assessment is used to calculate the losses caused by errors between the predicted and ground of truth data during training for backpropagation. In a hybrid model, the error loss caused by parameters is mainly calculated based on the overall objective function, guiding the optimizer to estimate the optimal parameters.

#### 3.2.2. Data Conversion

**3D point cloud to 2D depth map.** To ensure the consistency of features extracted by CNN networks, a point cloud is usually projected onto the camera plane as a 2D depth map for subsequent processing [35,44,45,46,48]. At this point, assuming the initial calibration parameters are $T_{\mathrm{init}}$, and the camera’s intrinsic parameters matrix are K, a depth map can be obtained by converting the point cloud $P^{L}$ according to Equation (3).

It is a fact that the depth map obtained by point cloud usually has few value points and many image blank holes. In order to avoid feature losses caused by this and ensure consistency with the final extracted feature map of RGB images, a pooling operation is usually performed after the depth map is input into the network to improve the density of the depth map, which is called the Dense Operation [38,44,48].

**2D RGB image to 2D depth map.** In some scenarios and studies [14,22,70], it is necessary to calibrate binocular or multiple cameras with LiDAR. The commonly used method is to convert the multi-camera image and LiDAR point cloud into disparity images [71] or depth maps [42,66] (which are essentially the same, and can be converted to each other through Equation (4)), thereby making the subsequent feature extraction and matching of network models more accurate. The multi-camera estimation of depth information has always been a key technology in the field of machine vision, with many mature methods including local feature-based methods [72,73], global feature-based methods [74], CNN model-based methods [75], etc. However, as this article focuses on extrinsic parameter calibration, the issue will not be discussed here. Among these algorithms, the semi-global matching (SGM)-based method [76] is one of the most widely used methods in practical applications from autonomous vehicle to automatic monitoring due to its computational efficiency, accuracy, and simplicity. By using the above method, the disparity can be obtained. By using Equation (4), the depth information can be calculated from the disparity, where B and f are the baseline and focal length of the stereo cameras, respectively.
(4)$$depth=\frac{B\cdot f}{disparity}$$

**2D image to 3D point cloud.** Cocheteux et al. proposed a network module in PseudoPillars [69] that directly converts 2D images to 3D point cloud. The 2D RGB image is processed to generate a depth map by global local path networks (GPLN) [77], and then the depth map is converted to 3D point cloud according to Equation (1), as shown in Figure 6.

**Spherical space transformation for 2D and 3D data.** Zhu et al. [78] argued that the camera projection parameters and point cloud density are key variables that affect generalization in the calibration task. These prior parameters will inevitably be incorporated into the inductive bias of the neural network during the training process. To address this problem, they proposed a generalization oriented data preprocessing strategy including the unified spherical space as shown in Figure 7, the data augmentation for camera images, and the set mapping process for LiDAR points.

#### 3.2.3. Feature Extraction

According to the architecture of the network, global feature extraction methods are divided into five categories: CNN, NiNs (Network in Network), ResNet, attention mechanism, and multi-resolution-based extraction. These methods are all based on the convolutional neural network.

**CNN-based feature extraction.** The CNN is the foundation and classic block of deep learning networks, and is also the most common method for image feature extraction [79]. NetCalib [66] uses a customized CNN for feature extraction. In the data conversion stage, this model converts both point clouds and RGB images into depth maps. The feature extraction network is mainly used for depth maps to preprocess the inputs and reduce their dimensionality.

**NiNs-based feature extraction.** The NiN (network in network) is a network structure proposed by Lin et al. [80]. A NiN block consists of a convolutional layer and a MLP, as shown in Figure 8. Due to the fast convergence advantage of the NiNs, both depth map and RGB image branches in RegNet [35] are processed by the NiNs to extract features. The RGB branch NiN block is pre-trained on ImageNet [81] and the depth map branches maintain symmetry with the RGB branches in architecture, but they reduce feature channels to reduce retraining. By adopting this architecture, the RGB branch ensures that the network is validated and reliable without long-term training processes. The depth branch reduces training, and it improves the efficiency.

**ResNet-based feature extraction.** ResNet [82] has powerful ability in image classification and is an ideal tool for image feature extraction. It is also the most commonly used feature extraction block in global feature-based methods [39,44,46,57,59]. Similar to the design in RegNet, the RGB branch uses pre-trained ResNet-18, while in the depth branch, the feature channels of ResNet-18 are halved for new training.

In these algorithms, the ResNet of different depths will be used to meet network design requirements. CalibNet [44], LCCNet [46], CALNet [39], CFNet [59], Calib-RCNN [57], CalibDNN [43], etc., all adopt ResNet-18, while RGGNet [45], CaLiCaNet [83] etc., use ResNet-50 to obtain more robust features.

**Attention mechanism-based feature extraction.** For the different imaging principles between the depth map and the RGB image, the correspondence between two kinds of features cannot be guaranteed. To improve the situation, LIA-SC-Net [38] proposed a method by the intensity attention mechanism. The basis for introducing this mechanism is the Lambert reflection model [84], which found that the object with higher LiDAR intensity has the potential of providing the salient co-observed feature in both laser data and optical images. The basic idea of this method is to generate an attention feature map through the laser intensity information, guiding the feature extraction and subsequent matching correspondence between the LiDAR and camera data, as shown in Figure 9a.

Instead, Shang et al. proposed another attention approach in CALNet [39], the channel attention mechanism [85]. In this mechanism, the feature map is globally averaged and pooled for each channel to obtain channel-level statistics. Then, the correlation between channels is captured through dimensionality reduction and activation layers. In this way, it enables the network to concentrate on channels and regions with high information content, as shown in Figure 9b.

**Multi-resolution feature extraction.** In order to preserve the description of features at different levels of resolution, improve the expressive ability of features, and enhance the accuracy of subsequent feature matching, Wu et al., inspired by HRNet [86], proposed PSNet [61], whose backbone network for feature extraction retained high-resolution representations and connected multiple resolution feature maps in parallel, as shown in Figure 10a. The advantage of this method is that the extraction of multi-resolution semantic representations is stronger than in existing methods. In addition, high-resolution channels can maintain more accurate spatial representation. 

In order to reduce semantic loss and improve processing efficiency, Liu et al. [87] proposed MRCNet, in which the multi-resolution processing is applied only to the last few layers of the RGB branch, as shown in Figure 10b. Instead, Xiao et al. [88] fuse multi-resolution features with different receptive fields, as shown in Figure 10c.

Similarly, Lv et al. proposed a multi-resolution feature extraction architecture in CFNet [59], which differs from the method in PSNet [61]. It extracts feature maps of different resolutions through a serial architecture instead of parallel connection and merging. The extracted feature maps are directly input into the subsequent serial feature-matching blocks of different resolutions for processing, as shown in Figure 11.

#### 3.2.4. Feature Matching

Feature matching is to obtain the corresponding relationship between points and pixels in both features. In an end-to-end mode, feature matching is only an intermediate process that does not directly provide corresponding results. Instead, feature maps of the two branches are aggregated into one data stream (feature map) through correlation, convolution, or connection, and the results are reflected in the feature map and the network’s weights. In mixed learning mode, some backbone networks will provide matching results, such as the CF given by CFNet [59] and DXQ-Net [41]. In some algorithms, there is no explicit matching (such as mutual information-based methods [89,90,91]), and in subsequent objective function optimization, both calibration parameters and correspondences are solved simultaneously. We summarize the network blocks used for feature matching in the following text.

**A.** 
**Feature matching based on optical flow estimation blocks**


The data processes for finding corresponding pairs of pixels between the images are the same with respect to optical flow and extrinsic calibration. Therefore, the extrinsic calibration network draws a lot on the ideas of optical flow networks.

**Feature matching based on NiNs or CBR.** After feature extraction from the two branches, RegNet [35] concatenates the feature maps, convolves them, and generates a joint representation, which is implemented through the NiN blocks. This approach is inspired by the correlation layers in FlowNet by Dosovitskiy et al. [92]. The correlation layer can perform multiplication comparison on two feature maps for correlation matching. The steps of correlation are the same as those of convolution, but in correlation the data are convolved with the other data, not the convolutional kernel. CalibDepth [47] adds a deformable convolutional network (DCN) [93] to the RegNet matching block to enhance the receptive field and its tolerance to different miscalibrations. Also inspired by RegNet [35], CalibNet [44], RGBDTCalibNet [48], the algorithm in paper [68], etc., it concatenates feature maps and convolves them to obtain feature correspondence relationships. However, the above methods are directly implemented through one or several levels of CBR (Conv + BatchNorm + ReLU).

**Multi-resolution correlation feature matching.** To improve the accuracy of feature matching, LCCNet [46] and CFNet [59], inspired by PWC-Net [60], adopt a multi-resolution serial architecture for feature matching. In CFNet, the two data branches’ feature maps are encoded at five different resolution encoders, and each is input into a concatenated decoder for feature matching. The initial matching results and feature maps at each level are obtained, and then input into the next decoder, as shown in Figure 11. Then they are connected to the context network, which is an extended convolutional network that can effectively expand the receptive field of the output unit at the pyramid level, producing accurate matching results. Duan et al. [94] also used this method in SCNet.

**Iterative matching based on pyramid 4D correlation volume (P4DCV).** Jing et al., inspired by the optical flow network RAFT [64], proposed DXQ-Net [41] and achieved feature map matching by constructing two blocks: pyramid 4D correlation volume calculation and calibration flow probability estimation blocks. The framework of the network is shown in Figure 12. There are three steps for calibration: Firstly, construct a feature 4D correlation pyramid through the dot product of RGB and depth feature maps. Secondly, referring to the optical flow probability regression technique [95,96], the optimal match is retrieved based on 4D correlation volume to perform CF probability estimation. Thirdly, the final CF estimation is obtained after 12 iterations.

**B.** 
**Feature Matching Based on Recurrent Network**


**LSTM-based feature matching.** In this type of method, it is considered that extrinsic parameters estimation is not only a spatial but also a temporal estimation problem, as shown in Figure 13a. Based on this, CalibRCNN [57] and some other methods [47,63] all use LSTM structure for matching, as shown in Figure 13b. During the matching process, the matching results at the previous time affect the current matching. 

**LTC-based feature matching.** Similar to the basic idea mentioned above, the CALNet [39] proposed by Shang et al. uses an HSPP combined with an LTC network [62] for feature matching. The advantage of the HSPP is the same as that of the SPP, which can adapt to and fully utilize datasets of different sizes and dimensions. LTC is a time continuous recurrent neural network model obtained by combining linear ordinary differential equation (ODE) neurons and special nonlinear weights, providing the same or even better performance than the LSTM model.

**C.** 
**Feature matching based on ResNet and DenseNet**


The basic idea of ResNet and DenseNet [97] is to connect the front layer to the back layer through a short cut to solve the problems of gradient vanishing, explosion, and network degradation. In extrinsic calibration, both of them are applied to feature matching.

**ResNet-based feature matching.** CalibDNN [43] proposes a matching architecture using ResNet, which connects front and back layers. In this algorithm, in order to reduce dimensions, half of the channels are used in the second block.

**Multi-resolution feature matching based on DenseNet.** In paper [62], Wu et al., inspired by reference [98], implemented feature matching for multi-resolution feature maps and merged multi-resolution information into features through cross feature-level fusion blocks. To achieve better matching results, the DenseNet module is used for further feature matching, as shown in Figure 14. The basic idea of the DenseNet model is consistent with ResNet, but it establishes dense connections (i.e., additive variable connections) between all previous layers and the following layers, as shown in Figure 15. This reduces the number of parameters and computational cost compared to ResNet, resulting in better results.

**D.** 
**Feature matching based on transformer**


Cocheteux et al. used the MobileViT (mobile-friendly vision transformer) [99] for feature matching in PseudoCal [69], as shown in Figure 16a. While in UniCal [100], a MobileViT network was used to achieve feature extraction and matching. MobileViT [99] is a lightweight universal vision transformer (ViT)-based on self-attention. It combines a convolutional neural network (inherent spatial bias induction and low sensitivity to data enhancement) and ViT (adaptive weighting and global information processing) to effectively encode local and global information and learn global representations from different perspectives. 

Xiao et al. [89] also employed a transformer architecture to obtain the correlation feature map for matching. In the algorithm, the query Q is derived from the initial pose query, while the key K and value V are obtained from the encoder’s output. They used a transformer decoder to extract further features for calibration parameters estimation, as shown in Figure 16b.

#### 3.2.5. Parameters Regression

In an end-to-end mode, the parameters regression refers to the estimation and prediction of calibration parameters. Unlike traditional calibration methods, the feature matching result is not given immediately after the matching stage. Therefore, in some networks, the boundary between feature matching and parameter regression is not clear. For a better explanation, in this paper, parameters regression is categorized into three types: MLP, CNN + MLP, and RNN + MLP. All other architectures are classified as feature matching. In hybrid learning mode, this step is called parameters optimization, which involves optimizing the objective function (or model) to obtain extrinsic parameters.

**MLP-based parameters regression.** The goal of the extrinsic calibration network is to predict the extrinsic parameters, and the most direct and classic regressing means is to use an MLP. RegNet [35] stacks two fully connected layers (FCs) for estimating dual quaternions. The last layers in RGGNet [45] and LCCNet [46] regress the rotation vector and translation vector through two FC branches. In order to adapt to any input size and achieve consistent output, Wu et al. added an SPP layer to the MLP for regression in NetCalib [66,101]. In addition, LIA-SC-Net [38] adopts a similar MLP regression architecture.

**CNN + MLP-based parameters regression.** In order to improve the receptive field and generalization ability of the network during global parameters regression, convolutional layers are used in networks such as CalibNet [44] and CalibDNN [43], in which two branches of FC are stacked to regress translation and rotation vectors.

**RNN + MLP-based parameters regression.** Aouragh et al. [102] applied an RNN + MLP architecture for regression. Before passing the data to the last fully connected layer, they concatenated it with a hidden state vector in order to have two parallel branches, i.e., containing output and a hidden state to be used as a recursive input like a recursive neural network (RNN).

**Traditional parameters estimation methods.** In hybrid learning mode, traditional parameters estimation methods are used in CFNet [59] and DXQ-Net [41]. After obtaining the 2D–3D data correspondence, CFNet converts the extrinsic calibration into a PnP problem, solved by the RANSAC + EPnP method [103]. In DXQ-Net [41], the problem is transformed as minimizing the semantic align model, as detailed in Section 3.3.5, and optimized using the Gaussian Newton method [104].

#### 3.2.6. Loss Function

The most common loss functions include the global calibration parameter loss function $L_{Para}$, point cloud reprojection loss function $L_{\mathrm{RePrj}}$, photometric loss function $L_{\mathrm{Photo}}$, structural consistency loss function $L_{\mathrm{Struc}}$, etc. Generally, loss functions can be used independently, but in order to impose stricter constraints on the network and accelerate its convergence, multiple losses are used simultaneously to describe the calibration problem. Finally, the loss function is constructed as follows:(5)$$L_{\mathrm{Total}}=\sum_{i}\alpha_{i}L_{i},\;i=\mathrm{Para},\mathrm{Project},\mathrm{Photo},\mathrm{Struc},\cdots$$

**A.** 
**Calibration parameter loss**


**Euclidean distance (ED).** The calibration parameter loss (CPL) is the error between the predicted rotation and translation parameters and the ground truths. The most straightforward metric is the Euclidean distance:(6)$$L_{Para}=L_{\mathrm{Tran}}+\lambda L_{\mathrm{Rot}}$$
where $L_{\mathrm{Tran}}$ represents the translation parameter estimation loss, $L_{\mathrm{Rot}}$ represents the rotation parameter estimation loss, and λ represents the regularization parameter. The Euclidean distance is used in models such as RegNet [35], CalibNet [44] and so on [39,43,57,62].

Translation parameters estimation loss. The most intuitive and common form for translation parameters loss is Euclidean distance with an L1 or L2 norm. This form is adopted in models such as RegNet [35], CalibRCNN [57], CalibDNN [43], etc.

Rotation parameters estimation loss. This includes quaternion Euclidean distance with an L1 or L2 norm [43,57], normalized quaternion Euclidean distance [105], and angular distance [46,62].

**Riemannian distance (RD).** The space defined by calibration extrinsic parameters has exceeded the scope of Euclidean space. Therefore, Yuan et al. [45] argued that using the Euclidean distance to define the error of parameters was inaccurate, and proposed RGGNet which used Riemannian geometric space. The distance between two points in a manifold is defined by the Riemannian metric, which is the inner product of the tangent space of the points on the manifold. Specifically, it is the geodesic distance. In DXQ-Net [41], the geodesic distance measurement is also used to describe the losses caused by global calibration parameter errors.

**B.** 
**Point cloud reprojection loss**


Point cloud reprojection loss is the error between the original and reprojected point clouds. The latter is reprojected from the depth map with ground-truth extrinsic parameters and the depth map is projected from the predicted extrinsic parameters (or real extrinsic parameters projection). This kind of loss function has been used in LCCNet [46], CALNet [39], and PSNet [61], etc. Kodaira et al. used a 3D Spatial Transformer Layer in CalibNet [44] to achieve the calculation of reprojection loss. Given two point clouds, they applied the following metrics:

**Centroid ICP Distance.** Similar to the loss term used for iterative closest point (ICP)-based alignment, they try to directly minimize the distance between the target point cloud and the point cloud transformed by the predicted transformation.

**Earth Mover’s Distance.** The earth mover’s distance is originally a measurement of dissimilarity between two multidimensional distributions. Since the distance between the points can be calculated, the earth mover’s distance is used as a metric that signifies the overall distance between the point clouds.

**Chamfer Distance.** The chamfer distance between two point clouds is defined as the sum of squared distances of the nearest points between the two clouds. Similar distances are also used in CalibDNN [43].

**C.** 
**Point cloud projection loss**


**Point projection Euclidean space loss.** For each 3D point $P_{i}$ in the point cloud, its predicted point in 2D is $p_{i,\mathrm{pre}}$, and the ground truth is $p_{i,\mathrm{gt}}$. This type of loss is defined as the Euclidean distance between $p_{i,\mathrm{pre}}$ and $p_{i,\mathrm{gt}}$, as that in RGKC-Net [106].

**Photometric loss.** This loss is defined as the dense pixel-wise error (since each pixel is encoded with the depth intensity) between the predicted and correct depth maps, as that in CalibNet [44]. CalibDepth [47] and CalibBD [63] also provide similar definitions, but with different names. In addition, CFNet [59] and DXQ-Net [41] define it as calibration flow. Specifically, CalibDepth uses a different metric called the berHu metric.

**Structural consistency loss.** Based on photometric loss, An et al. proposed a structural consistency loss function in LIA-SC-Net [38], which combines the intensity information of LiDAR point clouds. This function combines the laser intensity photometric and depth photometric losses to evaluate the overall structural consistency of point cloud data.

**Supervised Calibration Flow Loss.** In CFNet [59], Lv et al. proposed a supervised CF loss, which is similar to the photometric loss, and they defined it as the sparse pixel-wise error between the predicted calibration flow and the ground truth calibration flow.

**Calibration Flow Sparse Loss.** In CFNet [59], the calibration flow is sparse, and most of the pixels are invalid. Thus, the sparse loss is to enforce the displacement of pixels without ground truth to be similar to the displacement of the neighboring pixels. This sparse loss can also be used for point cloud depth maps and is applicable to all calibration models.

**D.** 
**Stereo constraint loss**


**Synthetic view constraint loss.** In CalibRCNN [57], the influence of time series is considered and the depth map is projected from point cloud with predicted extrinsic parameters. Based on the stereo correspondences between the depth map and RGB image, and changes in camera pose from time to time, a synthetic image can be generated. Synthetic view constraint loss is the photometric loss between the real image at the moment and the synthetic image.

**Epipolar geometry constraint loss.** Similarly, CalibRCNN [57] and CalibBD [63] also take the influence of time series into account. Changes in camera pose can be obtained from changes in point cloud with predicted extrinsic parameters. Thus, two camera images at adjacent times are obtained, and a loss function for epipolar geometric constraints is established.

**E.** 
**Supervised contrastive loss**


To enhance the convergence effect of positive and negative samples during training, Zhao et al. [70] introduced supervised comparison loss as follows:(7)$$L_{\mathrm{Sup}}=\sum_{i\in I}\frac{-1}{\left|P\left(i\right)\right|}\sum_{p\in P\left(i\right)}\log\frac{\exp\left(z_{i}z_{p}/\tau \right)}{\sum_{a\in A\left(i\right)}\exp\left(z_{i}z_{a}/\tau \right)}$$
where $P\left(i\right)\equiv \left\{p\in A\left(i\right):{\tilde{y}}_{p}={\tilde{y}}_{i}\right\}$ is the set of indices of all positives samples distinct from *i* within the mini-batch. $z_{i}$, $z_{a}$, and $z_{p}$ are features of the given negative and positive samples, respectively. τ is a scalar temperature parameter.

#### 3.2.7. Summary of Global Feature-Based Methods

This section comprehensively summarizes the existing deep learning-based global feature calibration methods from six aspects: data conversion, network data input, feature extraction, feature matching, parameters regression, and loss function, as shown in Table 1. From the table, it can be seen that most global feature-based methods use end-to-end learning networks. Due to the similarity between optical flow estimation, the design of extrinsic calibration networks is deeply influenced by existing optical flow networks, especially in feature extraction and matching blocks. These include correlation layers, multi-resolution matching, pyramid iteration matching, etc., which are based on FlowNet [92], PWC-Net [60], and Raft [65] optical flow networks, respectively.

From the perspective of network design and data processing, this type of method usually converts the point cloud into a depth map with the same dimension as the RGB image before feature extraction, so that the data consistency is ensured and the two data branches can be used for the networks with the same architecture. Thus, it can maintain global feature consistency in representation as much as possible, laying the foundation for subsequent matching and parameters regression. The ResNet network is the most frequently used feature extraction block, mainly due to its mature and stable performance. Moreover, a pre-trained model is provided. Feature matching is the most critical module of the entire network, and typically stacked by data-processing blocks such as CNN, ResNet, DesenNet, etc. As mentioned earlier, this does not provide specific feature correspondences but only a global feature map. From this point of view, this step can also be considered as a feature extracting step after aggregating the two features’ data. and it plays the same role for subsequent parameters regression. 

It is worth noting that there has been a trend to add an initial parameter correction block for the original data at the data conversion stage, such as the rotation alignment block [94]. 

Overall, we can obtain a basic model framework: At the data conversion stage, point clouds are projected into depth maps. The network inputs are depth maps and RGB images. At the feature extraction and matching stages, ResNet blocks and a CNN block are used, respectively. Finally, an MLP block is applied to perform the parameters regression. The loss function adopts the calibration parameters loss function.

### 3.3. Local Feature-Based Methods

A typical learning-based local feature extrinsic calibration is to segment or semantically encode camera RGB images and LiDAR point clouds based on deep learning networks, to obtain corresponding semantic objects or underlying features. Based on this, the initial estimation and fine optimization of extrinsic parameters are performed. Compared to global feature-based methods, local feature-based methods have certain advantages in terms of noise resistance, robustness, and adaptability to environmental changes, as they utilize local features. But there are certain requirements, and it requires certain structural features such as target objects [40], linear structures [108], etc., in shotting scenarios.

SOIC [36] is a semantic feature-based calibration algorithm proposed by Wang et al. in 2020. This algorithm applies pre-trained networks for the semantic segmentation of images and point clouds and it uses the semantic centroid (SC) of the segmentation results to estimate the initial rough pose. Based on the initial estimation, it optimizes the defined semantic consistency optimization model to accurately estimate the calibration parameters. In 2021, Peng et al. proposed the SemCal [32] method, which utilizes mutual information registration combined with the PnP method to solve the initial values of calibration parameters based on the semantic feature extraction network. Then, the mutual information neural estimator (MINE) [109] estimates the mutual information (MI) of the semantic features, and calculates accurate extrinsic parameters through gradient optimization. In SemAlign [50], Liu et al. defined a semantic alignment loss by minimizing out of class distance to quantitatively evaluate the calibration. Kodaira et al. [110] improved semantic alignment loss in SST-Calib and designed a bidirectional semantic alignment loss model for calibration optimization. The SE-Calib [58] proposed by Liao et al. in 2023 differs in features from the above methods, applying the semantic edge feature for calibration, and providing a semantic response consistency measurement model. The ATOP [40] deep learning calibration algorithm designed by Sun et al. in 2023 uses target-level feature matching. In the algorithm, a cross-modal matching network (CMON) is used to find the overlapping field of view between the camera and the LiDAR, and the 2D–3D target-level correspondences are calculated. In the optimization stage, particle swarm optimization is used to optimize the semantic alignment function to obtain extrinsic parameters between the LiDAR and the camera.

Zhu et al. [111] proposed a calibration method in 2020 that differs in strategy from the above approaches. This method does not extract semantic features from both the LiDAR and the camera data, but uses the pyramid scene parsing network [112] (PSPNet) to semantically segment the camera image. The segmentation results are then used as masks to guide the LiDAR point cloud to project onto the corresponding camera image. Thus, they constructed an optimization object and estimated the optimal parameters. The extracted semantic features are obstacles that can reflect laser beams, such as cars, trees, pedestrians, and traffic signs, and are used to construct optimization targets for optimal estimation. The Calib-Anything [52] calibration method proposed by Luo et al., inherits attributes of the above approach, but adopts the segment anything model (SAM) [113] for better semantic segmentation, which can adapt to more scenarios. It optimizes extrinsic parameters by maximizing the consistency of points projected within each image mask. Here, the consistency definition includes not only the position and the semantic information but also the intensity and the norm of points.

Aside from the aforementioned semantic-based methods, some scholars have also proposed algorithms that utilize more fundamental and low-level features. In 2022, Chao et al. proposed RGKCNet [106], a network based on 2D–3D key points. The algorithm adopts a deep declaration network to solve the calibration problem defined as bidirectional optimization, which can embed geometric constraint optimization into an end-to-end network and achieve 2D–3D data association. In 2021, Ma et al. proposed an automatic calibration and refinement method based on the line feature, CRLF [114]. They extracted line features in point clouds by a traditional method for line extraction, which extracted line features from lampposts and roads which are segmented by a network of RGB images, and they found the matching correspondence to estimate the parameters. In 2022, Hu et al. also proposed a calibration algorithm using line features, DedgeNet [108]. The difference is that the network extracts line features from RGB images, uses them as masks, and fuses them with the global feature map of the depth map. Then, the parameters matrix between the LiDAR and the camera is predicted. In 2023, Wang et al. proposed a similar calibration framework in FusionNet [115], but extracted key points from 3D point clouds instead of images. The 3D points were projected onto RGB images, and fused. Thus, the extrinsic parameter matrices were predicted in a coarse to fine estimation framework.

#### 3.3.1. Basic Framework

Usually, feature extraction-based methods are categorized according to the feature types. However, through research on existing deep learning-based local feature calibration methods, it has been found that there are two distinct data-processing frameworks within these methods, and the choice between these two frameworks is determined by whether the input LiDAR and camera data are performing local feature extraction simultaneously. If simultaneous (bilateral) local feature extractions are performed, the process is shown in Figure 17. If unilateral extraction is performed, the process is shown in Figure 18.

**A bilateral local feature extraction framework**, as shown in Figure 17, typically involves using deep learning networks to extract local features from both point clouds and RGB images, inputting the results into traditional calibration frameworks for matching, initial value estimation, parameters optimization, loss assessment, and so on. In a bilateral framework, since both data branches have undergone feature extraction, an initial estimation will be performed based on these features.

**A unilateral local feature calibration framework**, as shown in Figure 18, typically uses a network to extract local features from one of the two data branches. Another branch either does not perform any processing or performs global feature extraction. Then, the two data branches obtained are fused, and an optimization/regression is performed to obtain the parameters and evaluate the loss. In this framework, after a unilateral extraction of local features, the data streams of the two branches are usually fused, leaving both matching and parameter estimation for subsequent parameter regression/optimization steps.

**Feature extraction.** The typical step of local feature-based methods is to effectively segment 2D and 3D data through semantic segmentation networks, and then extract underlying local features based on semantic segmentation. In the process of semantic segmentation, one or more specific semantic objects are usually used as the target for feature extraction, such as pedestrians, vehicles, roads, etc. Low-level feature extraction refers to extracting features such as points, edges, and lines that can be used for matching on semantic objects or directly on images. Semantic segmentation and low-level feature extraction are not necessary steps for local feature extraction. They can be combined into one step, such as directly obtaining low-level local features from the semantic encoding part of a certain network, or directly optimizing parameters based on the results of semantic segmentation.

**Feature matching.** This step is a routine operation in feature-based calibration methods, mainly providing the correspondence between extracted features in point clouds and images. In the bilateral local feature calibration framework, the matching step after the semantic segmentation is omitted in most cases; for that, the corresponding relationship is established through semantic labels, or the extracted semantic objects are relatively few and do not require complex matching processes. In the unilateral local feature calibration framework, this step is usually replaced by the feature fusion step.

**Initial Guess.** Traditional parameters optimization models usually rely on initial values. When optimizing the objective function, in order to quickly achieve convergence and high-precision results, an initial guess of calibration parameters is usually carried out. The PnP (Perspective n Point) solving method is a commonly used method for extrinsic parameters estimation.

**Calibration model and optimization method.** The calibration model converts the extrinsic calibration of LiDAR and camera into a specific optimization objective function. Generally speaking, local feature-based methods will incorporate extrinsic parameters estimation and local feature matching into the optimization objective function, thereby simultaneously solving the two problems during the optimization process. Usually, the calibration model defined above is relatively complex and involves lots of parameters, and the objective function is non-convexity. Therefore, an optimization is needed to obtain an optimal solution.

#### 3.3.2. Feature Extraction and Matching

A.
**Bilateral local feature extraction**


**Semantic feature extraction and matching.** In 2021, Jiang et al. used the HRNet + OCR [116] and SalsaNext [117] for the semantic segmentation of 14 types of targets in point clouds and RGB images, respectively, in SemCal [32]. They applied mutual information metrics to semantic labels of different point–pixel pairs to establish overall correspondence. In the same year, Liu et al. proposed the SemAlign [50] method. By using SDCNet [118] for 2D RGB images and SPVNAS [119] for point clouds, feature extraction is performed separately, mainly considering semantic objects such as cars, roads, and sidewalks. Similarly, semantic labels are used to establish correspondences between objects. In 2022, Kodaira et al. improved the SemAlign method and proposed SST-Calib [110]. This method takes cars as the semantic object, and the RGB image processing is the same as SemAlign, but the point cloud processing adopts the more advanced SqueezeSeV3 [120] model.

**Semantic and semantic centroid feature extraction and matching.** In the SOIC [36] proposed by Wang et al. in 2020, PointRCNN [121] was used for point cloud semantic segmentation, and WideResNet38 [118] was used for image semantic segmentation. The two networks were pre-trained on the KITTI dataset, respectively. After semantic segmentation, the centroids of the RGB and point cloud semantic segmentation objects are obtained as feature sets for initial estimation and the matching correspondence between the two feature sets can be obtained from semantic labels.

In 2023, Sun et al. also used semantic objects’ centers and salient points in the ATOP algorithm [40]. However, different from SOIC, the ATOP requires point clouds to be projected to depth maps for LiDAR data. The two data branches input RGB images and depth maps to two semantic feature extraction networks (ERFNet [122] + MSHA [123]) with the same architecture to obtain target-level feature maps. ATOP calculates the matching correspondence between the target centers and salient points through a feature maps similarity calculation.

**Semantic edge extraction and matching.** In 2023, Liao et al. proposed SE-Calib [58] by extracting semantic edges as features. For the RGB image, the DFF [124] network is used to directly obtain semantic segmentation and extract edges, while the point cloud is semantically segmented by 3D-SS [125], and the semantic probability mapping and semantic label mapping of the point cloud are used to identify edge points and extract edge point features. The correspondence between the two is constructed based on the point cloud projection and semantic labels.

**Patch extraction and matching.** In 2023, Zhu et al. [78] extracted patches in feature maps as features. Firstly, the RGB image and the point cloud are transformed into a camera image and a LiDAR image. For the camera image, the EfficientDet [126] network is used to obtain the feature map, and for the LiDAR image the SqueezeSegV2 [127] is used. Then, feature maps are evenly divided into small patches as features. As patches of both images are obtained, two graphs are constructed. To obtain an explicit correlation matrix between cross-modal patches, the global graph embedding method is used to obtain the matching result.

**Straight line extraction and matching.** Ma et al. proposed a calibration method, CRLF [114], using straight lines as a feature, in 2021. In this method, the traditional RANSAC + line-fitting method is used for extracting lines in point clouds. The algorithm does not directly extract straight lines in RGB images, but uses the BiSeNet-V2 [128] semantic segmentation network to segment roads and light poles, thereby indirectly obtaining straight lines in the image. The matching between the two is carried out through Hough transformation.

**Key point extraction and matching.** In 2022, Ye et al. used key points for calibration in RGKCNet [106], as shown in Figure 19. The algorithm first extracts key points from point clouds and RGB images by human key point extraction networks [129] and PointNet++ [130], respectively. After key points extraction, the point weighting layer will further process point cloud features and output the weights of each point as well as the index of the top highest weights. Afterwards, through the feature embedding layer, the position, intensity, and local neighborhood features of the key point are extracted and embedded. A correlation correspondence generation layer is used for 3D and 2D points matching.

**B.** 
**Unilateral local feature extraction and fusion**


**2D image semantic mask.** Unlike the previous method, Zhu et al. only used the PSPNet [112] network to extract one semantic feature, vehicles, from RGB images in reference [111]. They used the vehicles to establish a reward mask. The fusion of 2D semantic features and 3D point clouds is relatively simple. The point cloud is projected onto the camera imaging plane, and the points falling into the mask are considered semantic correspondences. The framework of the algorithm is shown in Figure 20a. In paper [131], P. Root et al. used the Mask-CNN to obtain the mask.

Luo et al. used a similar mask strategy in Calib-Anything [52], but used the Segment Anything model [113] with better segmentation performance to segment multiple types of objects. At the same time, the position, intensity, and normal vector of the point are also used for fusion.

**2D line feature extraction and fusion.** Hu et al. [108] proposed an end-to-end calibration network based on the line feature. In this method, the line feature extraction is dominated by RGB images, and a DedgeNet convolutional network is constructed and trained to extract lines from images. The description of the line feature is the feature map encoded by DedgeNet. The point cloud processing branch first converts the point cloud into a depth map, and then uses ResNet-18 for encoding to obtain a global feature. The feature maps of two branches are fused based on correlation correspondence generation layers, as shown in Figure 20b. Due to the fact that the feature map encoded by DedgeNet describes a line feature, the output of the network is the result of the line matching result. This method can be seen as an architecture that introduces attention mechanisms, or as a unilateral local feature extraction architecture.

**3D point feature extraction and fusion.** In 2023, G. Wang et al. proposed a similar calibration framework in FusionNet [115], in which point features at different resolutions are directly extracted from the point cloud through PointNet++, and key points are projected onto the camera imaging plane to obtain the corresponding points in the RGB image and their descriptions in the global feature map, and fused. The parameters are predicted in a coarse to fine estimation framework, as shown in Figure 21.

#### 3.3.3. Initial Guess

Initial guess is an effective way for improving the convergence speed and accuracy of the traditional extrinsic parameters optimization. This section summarizes the initial guess methods used in existing learning-based local feature calibration methods.

A.
**Blind search without prior knowledge**


**Random sampling.** In SemAlign [50], Liu et al. randomly selected N transformation parameters, calculated their semantic alignment loss, retained the transformation parameters with the lowest semantic alignment loss, and used them for initial values. In the absence of prior conditions, random sampling is a feasible initial value estimation method, but this method is inefficient and requires a lot of attempts to obtain a good initial value.

**Brute force search + random sampling.** In order to improve the efficiency, Calib-Anything [52] used an exhaustive method combined with random sampling to estimate the initial values of calibration parameters. Firstly, use the exhaustive method to calibrate the rotation parameters in a large range with large step. Then, random sampling is applied to refine rotation and translation parameters within a small range.

**B.** 
**Camera pose estimation-based initial guess**


The aforementioned initial guess methods belong to a kind of blind search method with low efficiency. To improve the efficiency, existing feature information is usually used for initial value estimation.

**EPnP [103].** This is a non-iterative PnP algorithm, that represents the camera coordinates of reference points as the weighted sum of four control points, and then transforms the problem into solving the camera-frame coordinates of these four control points. This method is currently one of the most effective PnP solutions.

In SemCal [32], semantic labels are used to register RGB images with depth maps using mutual information-based methods, obtaining the initial pixels to points correspondence. Then, the EPnP method is used for initial value estimation. ATOP [40] uses a cross-modal matching network in the algorithm to match semantic targets, selecting the target center and its salient points as candidate corresponding point pairs, and performing initial guess by the EPnP method. In RGKCNet [106], proposed by Ye et al., the RANSAC + P3P [132] method used in paper [133] was introduced to initially estimate possible matching point pairs, and the initial extrinsic parameters were calculated for all possible matching point pairs through EPnP.

**IPPE [134] (Infinitesimal plane-based pose estimation).** IPPE can quickly and accurately calculate the pose from a single image of a planar object based on more than four pairs of point-to-point relationships. In SOIC [36], after obtaining centroids of semantic objects, the initial extrinsic parameters are calculated by the IPPE method.

**PnL [135] (Perspective n lines).** PnL is a problem of estimating camera pose using line features. The purpose is to calculate camera position and pose by known lines and their corresponding projections in the image. After establishing three or more line correspondences between the point cloud and the image, Ma et al. applied ASPnL [135] for initialization in CRLF [114].

#### 3.3.4. Calibration Optimal Model

Under the calibration framework, extrinsic parameters calibration requires constructing a specific optimization mathematical model based on the purpose and constraints of calibration. In end-to-end modes, it is usually represented as a loss function, which is described in Section 3.2.6. In hybrid learning mode, the mathematical model for local feature calibration is usually an objective function optimization problem related to point cloud 3D points, 2D image pixels, and calibration parameters, which can be expressed as follows:(8)$$\phi =\arg\underset{\phi }{\min/\max}D\left(P_{roject}\left(\phi ,K,P^{L}\right),p^{C}\right)$$
where D is the objective function defined based on different constraints or conditions, or the metric function defined between $P^{L}$ and $p^{C}$.

**Semantic response consistency metric model (SRCM).** SE-Calib [58] defines an SRCM model for semantic features. The SRCM counts the number of points in point cloud that correctly fall into semantic areas after projection, and the optimization model gets the extrinsic parameters by maximizing the SRCM. This model is the most direct and fundamental measure for the correctness of semantic points projected to the right semantic region of the image. This kind of model is also applied in ATOP [40].

**Semantic alignment loss model.** The optimization model is a minimum optimization model defined by the distance within the feature class, which constructs an objective function based on the distance between pixels and points of the same semantic class. Unlike the semantic response consistency model, the semantic alignment loss model incorporates distance measurement rather than just counts.

**Unidirectional alignment loss model.** In SemAlign [50], point clouds are projected onto a plane and a calibration optimization model minimizes the objective function which calculates the distance between projected points and pixels with the same semantic label. In Calib-Anything [52], the objective function measures the position, intensity, and normal vector of the point, more than just position.

**Bidirectional alignment loss model.** The unidirectional semantic alignment loss model is not so robust in complex scenarios and requires a good initial guess. SST-Calib [110] defines a bidirectional semantic alignment optimization model which can be better. The objective function not only calculates the distance from points to pixels, but also calculates the distance from pixels to points.

**Semantic projection consistency model.** SOIC [36] defines an optimization model, which is opposite to the SALM. The objective function sums the distance between pixels and points that are not of the same semantic class, and the distance uses the Manhattan distance. The optimization goal is to minimize the summation. 

**Semantic mask-based point cloud projection loss model**. For unilateral semantic feature extraction, Zhu et al. [111] proposed an optimization model using image semantic masks. In this optimization model, the objective function sums not only the distance of points correctly falling into the semantic region, but also the distance of points falling into the wrong region. The distance is defined using distance and inverse distance transformations [31].

**Semantic mutual information optimization model (SMIOM).** SemCal [32] considers the semantic label value of each point cloud and its corresponding image pixel as two random variables. The mutual information of the two variables should have the maximum value when they have the correct calibration parameters and the model uses the Donsker–Varadhan (DV) duality [109] to represent MI.

**Jensen–Shannon divergence-based loss.** While matching points by heatmaps, due to the symmetry of local numerical coordinates, the same matching result can have many possible heatmaps. In order to make the network converge to a specific position, the distribution of heatmaps needs to be regularized. RGKCNet [106] uses the Jensen–Shannon divergence to define this regularization. This function is a training loss function designed for an end-to-end network.

#### 3.3.5. Optimization Method

This section summarizes the optimization methods used in local feature calibration.

**Powell’s conjugate direction method [136].** This method was used to optimize the consistency function in SOIC [36]. It is a method of sequentially seeking the minimum points of unconstrained optimization problems along the conjugate direction. It has a superliner convergence rate, which to some extent overcomes the sawtooth phenomenon of the steepest descent method, while also avoiding the calculation and inversion problems of the Hessian matrix involved in the Newton method.

**Particle swarm optimization (PSO) [137].** PSO is a stochastic optimization technique based on swarm intelligence. PSO mimics the swarm behavior of insects, herds of animals, schools of birds, and schools of fish. These groups search for food in a cooperative manner, and each member of the group constantly changes its search mode by learning from its own experience and the experiences of other members. This method was used for optimization in ATOP [40].

**Gradient-descent method.** It is used in SemCal [32] to optimize both the MINE network weights and calibration parameters. Paper [111] adopts this descent method with annealing mechanism.

**Adam optimizer [138]**, used in SemAlign [50] and SST-Calib [110], has the advantages of simple implementation, high computational efficiency, and suitability for unstable objective functions, sparse gradients, or high noise gradients.

**Grid search [31].** In the process of solving the calibration parameters in SE-Calib [58], grid search optimization was used. This optimization method defines a parameter grid for each hyper parameter and its possible values, where the possible values of each hyper parameter are combined with those of other hyper parameters. Then, for each combination of hyper parameters, cross validation or other evaluation methods will be used to evaluate the performance of the model. Finally, select the hyper parameter combination with the best performance as the hyper parameter of the optimal model. This method is essentially an exhaustive search method, simple and intuitive. The disadvantage is that when the hyper parameter space is large, the search process becomes very time-consuming. To solve this problem, other optimization algorithms or techniques such as random search or Bayesian optimization can be used to accelerate the search process.

**Deep declaration network-based optimization.** This is an optimization network model used for end-to-end networks. In paper [106], RGKCNet describes the calibration problem as a bi-level optimization problem [41], which involves a global loss in the upper level and a point weight and matching observation loss in the lower level. From this, a declaration layer, PoseLayer, is constructed to solve the optimization problem of calibration. PoseLayer uses the limited memory Broyden Fletcher Goldfarb Shanno (L-BFGS) algorithm to estimate a six-degree of freedom pose. To ensure that the algorithm can converge to a better local optimum, the RANSAC method is first used to find the initial value.

#### 3.3.6. Summary of Local Feature-Based Methods

We summarize local feature-based methods from seven aspects: feature extraction structure, semantic object, feature description, feature matching, initial value estimation, calibration optimization model, and optimization method, as shown in Table 2.

In learning-based local feature calibration methods, the key lies in selecting appropriate point clouds and RGB images feature-extracting networks within the bilateral local feature extraction framework. Generally speaking, it is required that the features extracted by these two networks have classification correspondence. For example, the HRNet + OCR image semantic segmentation network in SemCal [32] can segment 20 classes of objects, while the SalsaNext point cloud semantic segmentation network can segment 16 classes. The intersection of the two can reach more than 14 classes, which can support the calibration. For example, in CRLF [114], the image semantic feature extraction network uses BiSeNet-V2 to effectively extract straight objects such as lanes and light poles, while point clouds use the traditional 3D line extraction RANSAC fitting method to construct corresponding features.

Under the framework of unilateral local feature extraction, the key to the algorithm lies in its calibration optimization. For example, Zhu et al. [111] defined a relatively complex calibration objective optimization function, which includes the errors between points falling inside the semantic mask and outside the mask in the calibration model. The semantic alignment loss optimization model of Calib-Anything [52] incorporates information such as the normal vector, intensity, and position of the point and it obtains preliminary estimates through a brute force search method, which sacrifices time and resources for accuracy.

From the perspective of algorithm structure, local feature-based calibration methods are mainly in a hybrid learning mode. The main reasons are as follow: Firstly, due to the differences in the structure of point clouds and RGB images, the encoded or extracted local feature data structure is inconsistent, making it impossible to input them into an existing learning network for parameter regression. However, adopting a hybrid learning mode can effectively complete calibration. Secondly, some traditional feature-based methods require more accurate and robust feature and semantic feature extraction networks (or underlying feature that constitutes a semantic object) to enhance the performance of algorithms.

With the continuous deepening of research, end-to-end calibration networks have also emerged in local feature calibration methods, such as RGKCNet [106], DedgeNet [108], and FustionNet [115]. DedgeNet projects the point cloud into a depth map and then inputs it, with the RGB image, into an end-to-end network. RGKCNet and Fusion-Net project the extracted 3D key points onto the image plane to obtain the same position points on the image plane, intercept the feature tensors that are consistent with the 3D point feature tensor structure, and then input the two into the subsequent matching layer. Regardless of which method is used, it involves a process of data conversion. DedgeNet uses the raw data before network input, while RGKCNet and FusionNet use the feature data in the network. Both of these data conversion methods are feasible, but undoubtedly the design of the latter is a more ideal end-to-end network structure.

In addition, it should be noted that in semantic feature-based calibration methods, a semantic feature usually plays two roles: The first is to roughly establish the correspondences between point clouds and RGB images, thereby perform an initial guess. The second is to obtain semantic regions and labels, and optimize and estimate the overall parameters using semantic labels as point and pixel attributes in different roughly matched semantic regions. From this perspective, semantic feature-based methods can also be seen as regional global feature calibration methods.

### 3.4. Relative Extrinsic Parameters Prediction

Relative prediction refers to the method of constructing a historical calibrated gallery set, retrieving historical data using current observational data, and predicting parameters based on the correlation between observational data and retrieval result.

In 2020, Wu et al. proposed a calibration network with a completely different approach from previous studies, CalibRank [37]. Inspired by the research findings on the interpretability of neural networks in references [139,140], which show that “pose regression is more closely related to pose approximation via retrieval than to accurate pose estimation via 3D structure”, this method does not solve the extrinsic calibration problem directly by capturing the correspondence between point clouds and RGB images, but estimates relative pose. CalibRank retrieves a historical calibrated gallery set using the current data as query data, gets the retrieval result, and fuses the retrieval result according to the relevance between query data and retrieval data, to obtain the relative calibration parameters, as shown in Figure 22.

This method is inspired by the RPR (relative camera pose regression) [141,142,143,144] methods using deep learning in visual localization problems [9,141,145]. The problem of visual localization is to accurately estimate the camera pose in a known scene, while relative camera pose regression is to retrieve relevant training images for predicting test images, and estimate the camera pose by comparing the relative poses with these images. Typically, this method can use explicit image retrieval steps [143,144] or implicitly represent images in CNN [142] to find relevant training images. Compared with absolute camera pose regression (APR) that estimates pose directly through image features [140] and needs to be trained for specific scenes, RPR is a more general problem which can be trained on multiple scenes.

As shown in Figure 22, CalibRank [37] retrieves the first relevant images and their calibration parameters from the calibrated gallery set, estimates the relevance between query data and retrieval data, and fuses them to obtain the relative parameters. This method introduces a learning–sorting process to sort the first relevant poses in the training gallery set, and then integrates them into the final prediction. In order to better explore the pose relevance between ground truth samples, an exponential mapping from parameters space to relevance space is further proposed. This method is divided into four parts: data input, data retrieval, relevance mapping, and pose fusion.

#### 3.4.1. Data Input

The system’s input I includes query sample q and the calibrated samples gallery set G with their own extrinsic parameters.
(9)I={q,G},
where *q* refers to tuples composed of a projected depth map and an RGB image.

#### 3.4.2. Data Retrieval

In data retrieval, the network needs to predict the relevance rel between the query sample q and each sample in G, and select the most relevant n samples from G, labeled as $S_{i},i=1,2,\cdots ,n$. The extrinsic parameters corresponding to each $S_{i}$ are $\phi_{i}$, and in the later fusion part, the extrinsic parameters $\phi_{n}$ of $S_{n}$ are fused to obtain the final prediction. 

Given a query q, the retrieval aims to retrieve $S_{n}$ from G. This section contains two modules, encoder E and sorter R. Encoder E first maps each sample in I to the embedding space.
(10)v=E(I|ω)
where v represents the embedding vector, and ω is the parameter of the encoder. Then, the connection layer connects the query embedding vector $v_{q}$ with each embedding vector $v_{G}$. Through this step, each combination vector $v_{c}$ has the combination features of q and G samples.
(11)$$v_{c}=\mathrm{concat}\left(v_{G},v_{q}\right)$$

Input $v_{c}$ into the sorter R to predict the correlation rel between q and G. σ is obtained by sorting rel, and $S_{n}$ are the first n samples of σ.
(12)$$rel=R\left(v_{c}|\omega_{r}\right),\sigma =\mathrm{sort}\left(rel\right),S_{n}={\mathrm{top}}_{n}\left(\sigma \right)$$

#### 3.4.3. Relevance Mapping

During the training sorting process, correlation should be provided as a true value. Due to the fact that the correlation between query samples and library samples cannot be directly obtained from extrinsic parameters, parameters distances need to be mapped into the relevance space. CalibRank [37] defines the relevance as a metric of the parametric distance δ in the same extrinsic parameters (e.g., yaw angle), and the relevance should be negatively correlated with δ. Moreover, when δ is less than a threshold, the difference in relevance between any two samples should be as big as possible. Therefore, it uses an exponential function to map the correlation, i.e.,
(13)$$rel=e^{k\delta +b},b=\log t-kM,$$
where M is the max range in the solution space, t is the minimum value of relevance, and k is the parameter set artificially.

#### 3.4.4. Late Fusion

The late fusion part aims to fuse $\phi_{n}$ into the final prediction $\phi_{\mathrm{pre}}$. Since the more relevant sample should weigh more in the final prediction, they weight $\phi_{n}$ with the corresponding normalized relevance and then sum them up as the final prediction.
(14)$$\phi_{\mathrm{pre}}=\sum_{i=1}^{n}\phi_{n}\cdot \mathrm{norm}\left(rel_{n}\right)$$
where $\phi_{\mathrm{pre}}$ denotes the extrinsic parameters predicted by the method.

#### 3.4.5. Summary of REPP

There has been a lot of research on relative camera pose regression methods in the field of visual location, and they have achieved performance similar to that of absolute camera pose regression methods. However, this kind of method, referred to as relative extrinsic parameters prediction, is still a novel concept for the calibration of LiDAR and camera. In principle, there is a similarity between visual localization and the extrinsic calibration of LiDAR and camera, that is, both use stereo vision technology to estimate the pose of sensors. The difference is that the scenes for a visual localization are actually limited, and the historical image data with known poses can basically contain key scenery. When querying, the observational data are contained in these sceneries, but for extrinsic calibration, the scenes are infinite, which means that the current observational data and historical data, ostensibly, may not have any relevance. Therefore, the theoretical basis for whether REPP is feasible, is still not solid enough at present.

For this reason, currently only CalibRank in reference [37] has adopted this method to solve the calibration problem. Based solely on the experiments and results provided in this reference, it has achieved calibration results similar to the CalibNet method, which brings a glimmer of hope for the research of this type of method. However, due to the limited experimental data and incomplete comparison, it cannot be fully demonstrated that CalibRank has the same performance as the state of the art methods such as LCCNet [46] and CalibDNN [43].

Objectively speaking, we have some comments about REPP. Firstly, it provides a different approach for extrinsic calibration compared to previous methods. Secondly, putting aside the data representation, the principle of estimating sensor pose using stereo vision technology is consistent between the visual localization and the calibration. Thirdly, there should be an inherent relationship between the point cloud and image data and the external parameters. However, this aspect has not yet been thoroughly studied by scholars.

## 4. Dataset

We summarize the commonly used datasets for deep learning-based extrinsic calibration networks, including the following:

**ImageNet [81].** This dataset is a large visualization database used for visual object recognition software research, with over 14 million manually annotated images. Corresponding bounding boxes are provided in at least one million images at the same time. This database was first published by researchers from Princeton University at the computer vision and pattern recognition (CVPR) conference held in Florida in 2009. In research on extrinsic calibration, this dataset is mainly used for training and pre-training network models for two-dimensional RGB image data stream processing, such as RegNet [35], CalibNet [44], RGGNet [45], RGBDTCalibNet [48], LCCNet [46], CalibDNN [43], CalibDepth [47], CFNet [59], DXQ-Net [41], etc.

**Cityscapes [146].** This dataset focuses on the semantic understanding of urban street scenes, and is a benchmark suite and a large-scale image and video dataset for training and testing pixel-level and instance-level semantic labeling methods. The urban landscape in the dataset consists of a large and diverse three-dimensional video sequence recorded on the streets of 50 different cities. There are 5000 high-quality pixel-level finely annotated images in the dataset and 20,000 roughly annotated images which can be used for a large number of weakly labeled data methods. This dataset was used for training and testing image semantic segmentation network models in algorithms such as SOIC [36], SemAlign [50], and SE-Calib [58].

**KITTI [147].** This dataset is jointly founded by the Karlsruhe Institute of Technology in Germany and the Toyota American Institute of Technology. It is currently the largest computer vision algorithm evaluation dataset in the world for autonomous driving scenarios. It contains multiple mode data (such as RGB images, grayscale stereo image pairs, 3D point clouds, GPS/IMU, etc.), which can be used for robot and autonomous driving research. KITTI records a 39.2 km visual ranging sequence and over 200 k 3D annotated object data, including detection, mileage, depth map, semantic, and other sub datasets, which can be used to evaluate the performance of computer vision technologies such as stereo images, optical flow, visual ranging, 3D object detection, 3D tracking, calibration, etc. in vehicle environments. In existing deep learning-based calibration, most of the methods use this dataset as the training and testing set for RGB image and point cloud processing networks, as well as for the entire calibration algorithm, such as RegNet [35], CalibNet [44], CALNet [39], SE-Calib [58], FusionNet [115], CalibRank [37], etc.

**KITTI-360 [148].** This dataset is an upgrade of KITTI, recording a driving distance of 73.7 km, with more than 320 k images and 100 k LiDAR scans. The dataset has been annotated with static and dynamic 3D scene elements, and has dense semantic and instance annotations for 3D point clouds and 2D images. It also has 19 categories available for evaluation, and its semantic label definitions are consistent with the Cityscapes dataset. This dataset was used for training and testing in RGBDTCalibNet [48], CFNet [59], DXQ-Net [41], SemCal [32], and so on.

**RELLIS-3D [149].** This dataset is a multimodal off-road environment dataset collected at Texas A&M University’s Raleigh campus, containing 13,556 frames of LiDAR scanning data and 6235 image annotations. This data can test the algorithm’s dependence on environment and categories, improving the algorithm’s robustness. CalibDNN [43], paper [70], and SemCal [32] have applied this dataset for testing.

**Carla Simulator [150].** In SemCal [32], in order to test algorithm performance, in addition to using real datasets, an autonomous driving simulator was also used to generate simulation test datasets. The Carla Simulator is an open source simulator for automatic driving research, which can support the most basic development, training, and verification of an urban auto drive system. The simulator can flexibly configure sensing kits and environmental conditions, generate simulated LiDAR and camera data, and meet the data testing requirements in extrinsic calibration.

## 5. Calibration Effects

We compare the calibration effects of different methods in Table 3. The calibration data are from the same dataset of KITTI, and the mis-calibrated range varies from [−0.1 m, 0.1 m]/[−1°, 1°] to [−1.5 m, 1.5 m]/[−20°, 20°]. The larger the range, the greater the difficulty of calibration, and the larger the error that may occur during calibration. 

In Table 3, we can find that although the mis-calibrated range is [−1.5 m, 1.5 m]/[−20°, 20°], CFNet [59] gave a great calibration result with a mean translation absolute error (TAE) of 0.995 cm and a mean rotation absolute error (RAE) of 0.087°. Figure 23 shows the calibration results of CFNet. 

PseudoCal [69] also performed very well with an RAE of 0.05°. Figure 24 shows the calibration results of PseudoCal. 

Ignoring the mis-calibrated range factor, we find that Z. Zhang’s method [70] demonstrated a mean TAE of 0.33 cm and PseudoCal [69] gave a mean RAE of 0.05°. DedgeNet [108] and CalibDB [63] also win the single TAE or RAE competition, respectively. From the results data, it can be seen that LCCNet [46], SCNet [94], and ATOP [40] also had good performance.

We also compared the robustness of the algorithms. A simple evaluation of robustness is conducted from the following three aspects: (a)In the experiment, were different types of datasets or real-time data used for testing? If two or more different types of datasets or real-time data are used, a score of two stars (☆☆) will be determined.(b)In the experiment, was the initial error range of the test data large enough? If error range ≥ [−1.5 m, 1.5 m]/[−20°, 20°], then one star (☆) is obtained.(c)In the experiment, was an anti-noise test is conducted? If so, one star (☆) will be awarded.

Under this evaluation standard, the maximum score is four stars. In Table 3, there is no algorithm reaching three stars and above, and there are only several algorithms with two stars which are tested with different datasets. It is worth noting that there is no work on anti-noise testing. We cannot guarantee that this standard can accurately evaluate the robustness of the algorithm, but at least it can demonstrate that there is still a lot of work to be conducted to prove the robustness of the algorithm. 

Overall, we find some interesting phenomena, as follows:(a)Methods of the AEPE performed better than the method of the REPP.(b)For the AEPE, the global feature-based methods generally have better calibration effects than the local feature-based methods.(c)End-to-end methods generally calibrated better than hybrid methods.

## 6. Discussion

In this paper, we provide a systematic overview of deep learning-based extrinsic calibration methods for LiDAR and camera. The current learning-based calibration methods can be divided into two categories, namely accurate extrinsic parameters estimation and relative extrinsic parameters prediction methods. The AEPE methods are currently the mainstream research direction, which can be divided into global feature-based methods and local feature-based methods. Relative prediction methods have more research in visual localization, and only preliminary research is currently conducted in the LiDAR and camera extrinsic calibration.

The AEPE method is to accurately estimate the extrinsic parameters based on the current observational data. The specific method based on global features is to extract global features from LiDAR-projected depth maps and camera RGB images through deep learning blocks. The network matches global features of the two as a whole, and regresses the predicted parameters. The advantage of this type of method is that it has low dependency on the scene and can be used in environments without obvious structures or targets. Without the need for special targets or any geometric constraints, it can achieve multimodal data alignment. Shortcomings: Firstly, the generalization ability of common problems in deep learning is relatively weak. Secondly, the interpretability is poor, especially for end-to-end networks. Finally, due to the lack of pre-trained network models for depth maps, it takes time to train existing models for depth maps. In addition, special depth map datasets have limited resources.

The local feature-based methods refer to the use of learning networks to extract local features of point clouds and RGB images for parameters estimation. Among them, local features typically include semantic targets [36,40], semantic regions [32], key points [106,115], edges [108], etc. According to feature extraction structure, we divided them into bilateral local feature extraction- and unilateral feature extraction-based methods. A bilateral local feature extraction-based method is a conventional data-processing structure in calibration, which involves extracting the same local features from both the camera and the LiDAR data, and then matching features to estimate parameters. However, due to the fact that point clouds and optical images often capture different features of the environment, they are easily affected by random factors such as noise and occlusion [111], and it is difficult to fully maintain consistency in extracted features. To overcome the above problems, a unilateral local feature extraction-based method has emerged, which involves only extracting local features on a two-dimensional or three-dimensional data stream, while the other data stream only performs simple data processing or even no processing. Through the fusion of the two data streams and subsequent optimization, they can obtain the parameters.

For local features, most existing methods use semantic features, and a small amount of them use geometric or other features. The main reason is that existing deep learning models tend to complete high-level semantic-related tasks. However, geometry and other low-level local features, to some extent, perform more accurately in space, have stronger generalization ability, even deviate from specific scene structures, and are more interpretable. Therefore, research in this area can provide broad application prospects, and is a potential research direction.

The advantages of this type of method are as follows: Firstly, the input data generally does not require complex preparation, and can be directly input as a 2D RGB image and 3D point cloud. Secondly, due to the extensive and mature research on learning networks for 2D RGB image and 3D point cloud, this type of method can directly use the corresponding pre-trained network model without training. Thirdly, mature pre-trained network models typically have good generalization ability through a large amount of data. Choosing a mature local feature extraction network model can effectively improve the generalization ability of such algorithms. The disadvantage of this type of algorithm is that it relies on local features and requires the scene to have a certain structure or semantic targets.

The REPP is a method of predicting and estimating the observational data through the historical data. Compared with instance-based AEPE, it is a more general problem with better generalization ability and is more likely to be used in various scenarios. This method is inspired by the visual localization method, and there is only one relevant paper available for LiDAR and camera extrinsic calibration, and the corresponding research is still in its early stage. Due to the differences between visual localization and extrinsic calibration, there are still many issues worth exploring. For example, the construction of a historical calibrated gallery set, whether this dataset is constructed from raw data or its inherent driving factors, is a question worth discussing. If constructed through internal driving factors, what are the internal driving factors that are still open? For example, in visual localization, data retrieval is based on the similarity criteria between observational image and historical image sets. While in calibration, it involves two types of data, and their observational data and historical data may be completely different and have no relevance. How should the retrieval criteria between them be defined?

## 7. Conclusions

LiDAR and camera extrinsic calibration are the foundation and prerequisite for unmanned intelligent perception systems to integrate data and perceive the surrounding environment. The method based on deep learning is one of the important methods that have emerged in targetless calibration in recent years. It can more efficiently and accurately obtain information from the surrounding environment in the data, and achieve calibration. Although the time for studying this issue is relatively short, gratifying results have been achieved. We believe that as research in this field continues to deepen, deep learning will play a unique role, just like in other visual tasks. In future research, the deep learning-based extrinsic calibration will face the following challenges which are worth delving into:

Although the research is still in its early stage, existing methods can still be clearly divided into two categories: AEPE and REPP. As mentioned earlier, an REPP-based method uses a historical retrieval and prediction method, which has significant differences from previous calibration schemes, but has also achieved certain results [37]. Due to limited research and many unresolved issues, as discussed in Section 6, further research is warranted.

AEPE-based methods are based on instance observational data, with relatively more research and excellent calibration results [46], and practical applications [84]. However, compared to the traditional targetless calibration methods, methods and types of networks used for calibration are still limited, mainly focusing on feature-based methods, especially for global feature extraction methods. Only a small amount of research has been conducted based on local features, and the ego motion method [151,152] has not been involved. In addition, the global feature extraction methods mostly use ResNet, and are of the simplex variety. From this perspective, there is still significant research space in this field.

From the perspective of the learning-based algorithm design, the end-to-end network method is undoubtedly a more perfect algorithm structure. However, most existing end-to-end networks rely on 2D image processing learning networks. Due to the differences in imaging principles between depth maps and RGB images, there is unpredictability in feature extraction and matching, which can easily lead to failure in actual scene calibration. The method of directly processing 2D and 3D data is more in line with natural laws and human information processing processes, but most of these methods are limited by the existing network design level and adopt a hybrid learning mode. Fortunately, in the past year of development, there have been a few end-to-end networks that directly process point clouds and RGB images, such as RGKCNet [107] and FusionNet [115]. Therefore, the design of calibration networks that directly process two (or even multiple) different dimensions and properties of data will be a research trend and a challenging issue in this field.

In practical large-scale application scenarios, such as autonomous driving and intelligent monitoring, low resolution LiDARs are often used due to cost and other reasons. In this case, extrinsic calibration is a very challenging problem, and it is one of the most urgent problems to be solved in practical applications and technology promotion. In addition, most existing studies are based on relatively ideal urban road datasets (primarily the KITTI dataset), prioritizing the pursuit of algorithmic precision. However, in real-time, more complex environments, influenced by factors such as noise and environmental shifts, enhancing the robustness and adaptability of algorithms is also a pressing unsolved problem for practical applications. Unfortunately, the research on these issues in deep learning-based calibration methods is still far from sufficient and needs further strengthening.

## Figures and Tables

**Figure 1 sensors-24-03878-f001:**
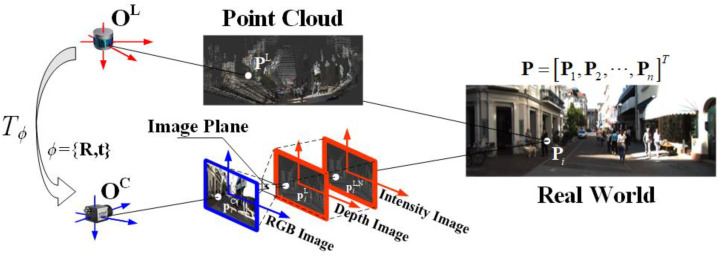
Coordinate system of LiDAR and camera and data transformation. The RGB image, the depth map, and the intensity map are actually overlapping with each other on the imaging plane. We plot them in a staggered manner to distinguish them.

**Figure 2 sensors-24-03878-f002:**
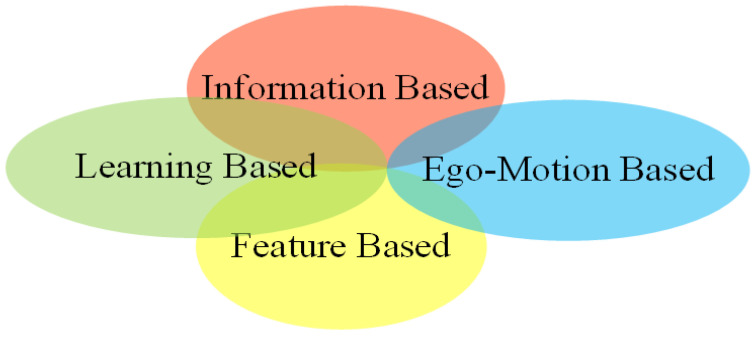
Classification of targetless calibration methods in reference [4].

**Figure 3 sensors-24-03878-f003:**
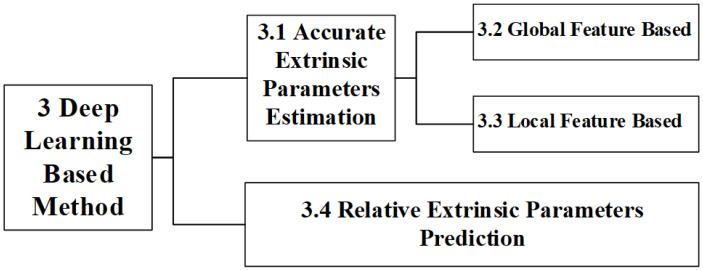
Classification of learning-based extrinsic parameter calibration method.

**Figure 4 sensors-24-03878-f004:**
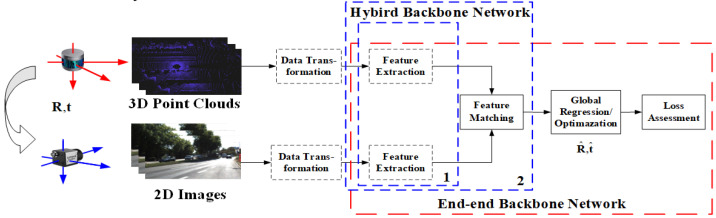
Basic pipeline of learning based calibration. The coordinate system formed by the red axis is the LiDAR coordinate system, while the blue axis is the camera coordinate system. The following figures in this paper are all represented in this way. The dashed blue box labeled 1 is a feature extraction network, and The dashed blue box labeled 2 is a feature extraction and matching network.

**Figure 5 sensors-24-03878-f005:**
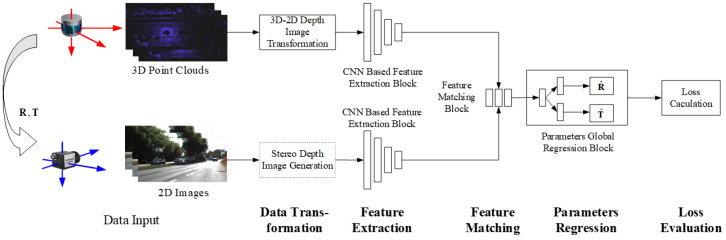
Pipeline of global feature-based calibration.

**Figure 6 sensors-24-03878-f006:**
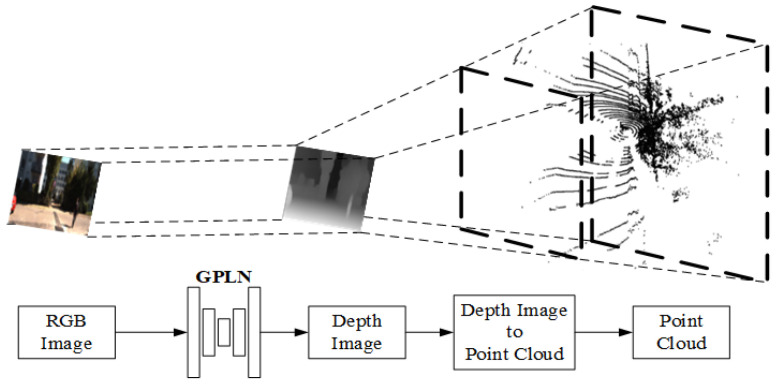
Data transformation from 2D RGB image to 3D point cloud.

**Figure 7 sensors-24-03878-f007:**
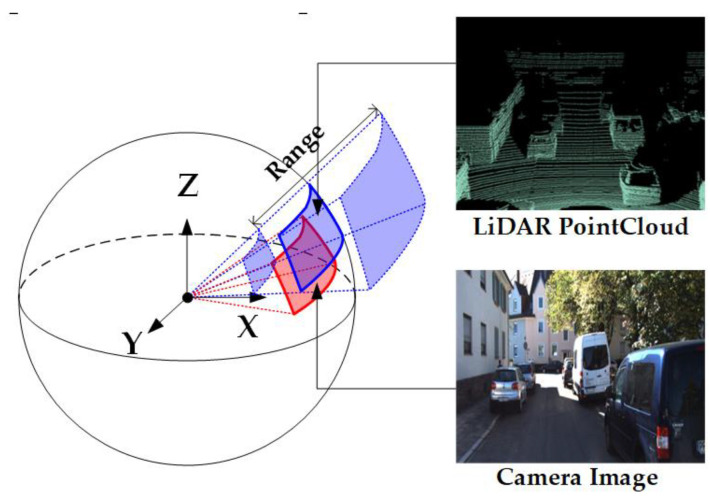
Spherical space transformation for 2D and 3D data. The blue solid box denotes the transformed spherical image for the 3D point cloud. Range in the figure denotes the points’ range in the point cloud. The red solid box denotes the transformed spherical image for the RGB image.

**Figure 8 sensors-24-03878-f008:**
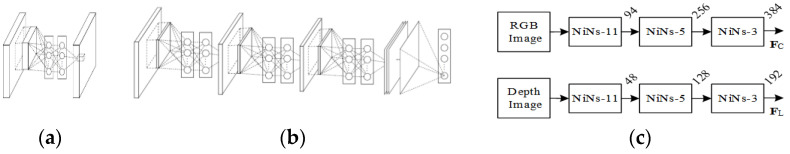
Architecture of NiNs. (**a**) Mlp + Conv-based, (**b**) NiNs-3-based, and (**c**) NiNs-based feature extraction.

**Figure 9 sensors-24-03878-f009:**
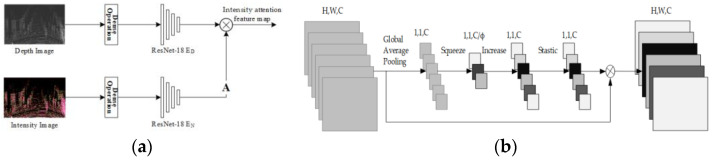
Feature extraction based on attention: (**a**) intensity attention mechanism and (**b**) channel attention mechanism.

**Figure 10 sensors-24-03878-f010:**
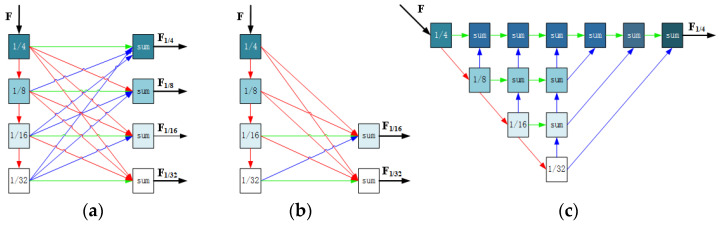
Multi-resolution feature extraction. (**a**) Multi-resolution representations; (**b**) multi-resolution representations with fewer layers; and (**c**) fused multi-resolution features. In the figure, the red line indicates downsampling, the blue line indicates upsampling, and the green line indicates no sampling.

**Figure 11 sensors-24-03878-f011:**
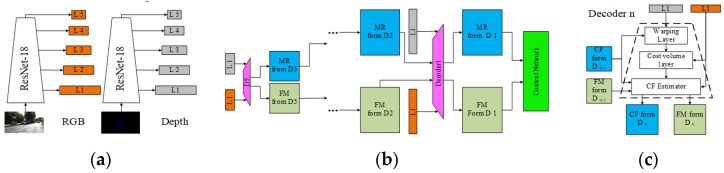
Multi-resolution feature extraction and matching strategy for CFNet. (**a**) Multi-resolution feature extraction; (**b**) multi-resolution feature matching, D n denotes Decoder n; and (**c**) Decoder n.

**Figure 12 sensors-24-03878-f012:**
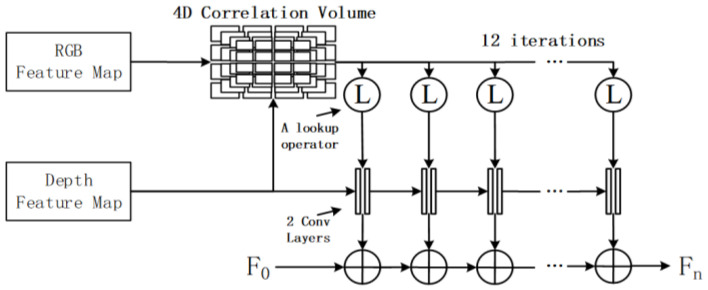
Iterative matching solution based on 4D correlation volume.

**Figure 13 sensors-24-03878-f013:**
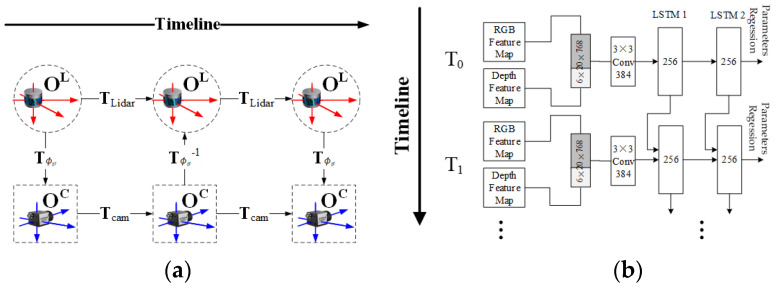
LSTM-based feature matching: (**a**) calibration on time sequences and (**b**) LSTM-based feature matching.

**Figure 14 sensors-24-03878-f014:**
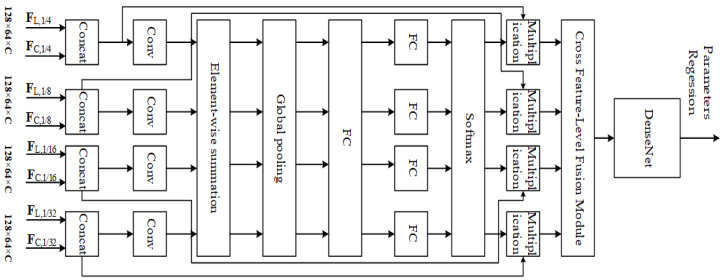
Multi-resolution feature matching in PSNet.

**Figure 15 sensors-24-03878-f015:**
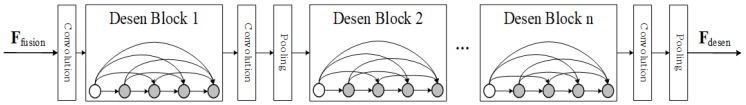
Architecture of DenseNet.

**Figure 16 sensors-24-03878-f016:**
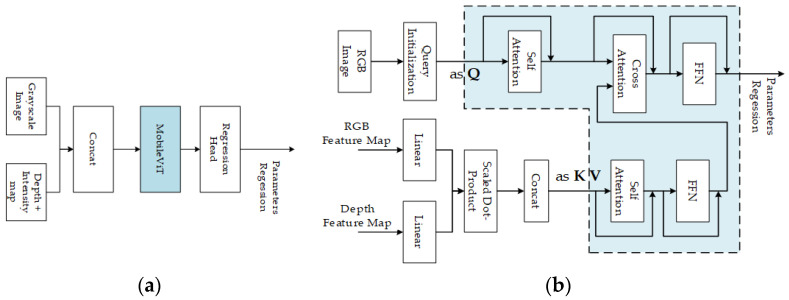
Feature matching based on transformer. (**a**) MobileViT-based feature matching; (**b**) Transformer architecture-based feature matching. In the blue box, the transformer is applied for matching.

**Figure 17 sensors-24-03878-f017:**
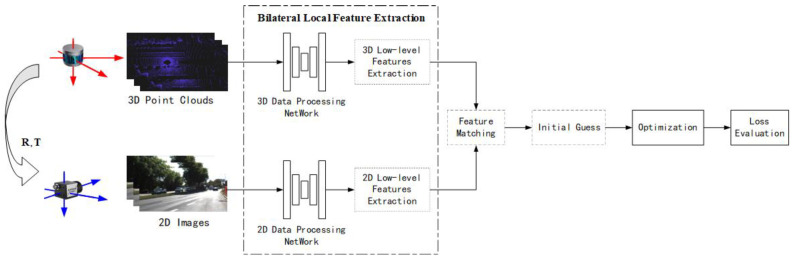
Bilateral local feature extraction framework.

**Figure 18 sensors-24-03878-f018:**
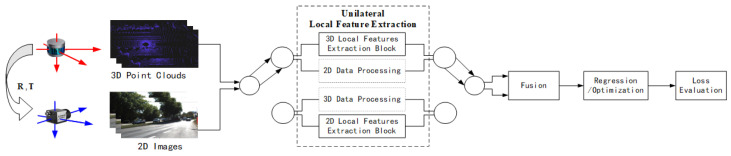
Unilateral local feature extraction framework.

**Figure 19 sensors-24-03878-f019:**
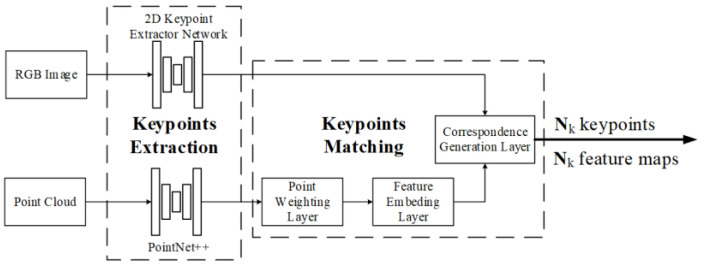
Key point extraction and matching.

**Figure 20 sensors-24-03878-f020:**
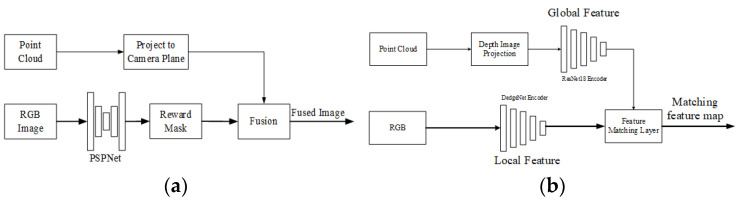
Unilateral semantic feature and line feature extraction. (**a**) Semantic feature mask extraction framework. (**b**) Unilateral line feature extraction and fusion.

**Figure 21 sensors-24-03878-f021:**
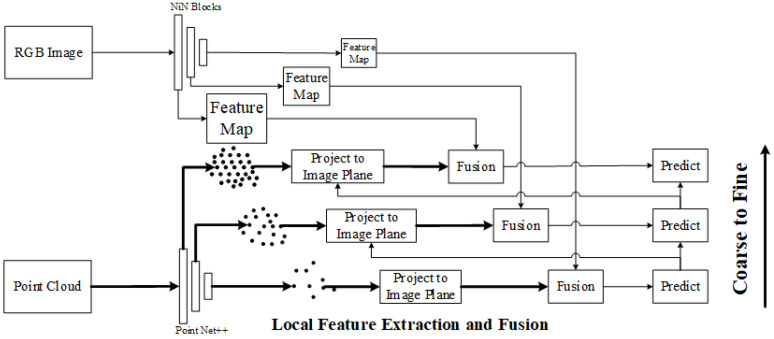
Unilateral semantic feature and line feature extraction.

**Figure 22 sensors-24-03878-f022:**
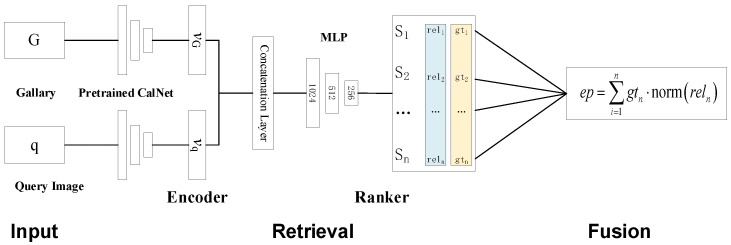
Relative calibration prediction.

**Figure 23 sensors-24-03878-f023:**
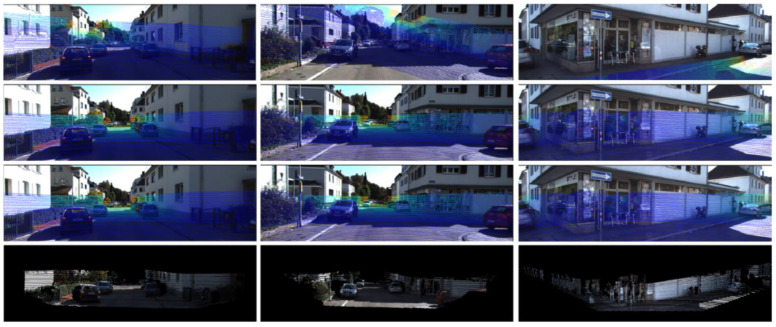
Visual calibration results of CFNet. Rows represent the initial miscalibrations, CFNet’s calibrations, and ground truth. The last row shows the colorized point cloud with the predicted extrinsic parameters.

**Figure 24 sensors-24-03878-f024:**
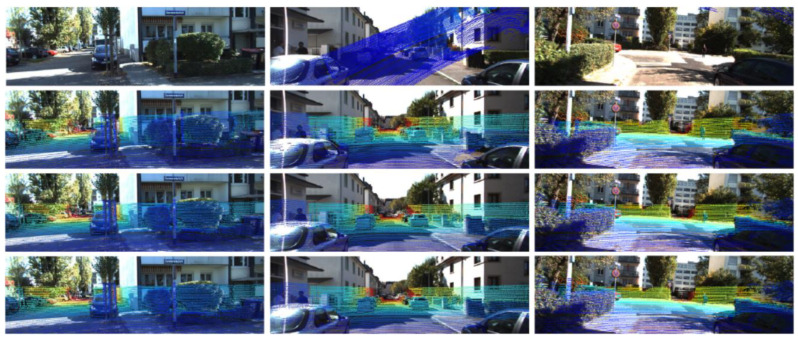
Visual calibration results of PseudoCal. Rows represent the initial miscalibrations, coarse adjustments, final refined calibrations, and ground truth.

**Table 1 sensors-24-03878-t001:** Learning-based global feature calibration methods.

Method, Year	Mode	Data Conversion	Network Input	Feature Extraction for Camera Branch, Feature Extraction for LiDAR Branch	Feature Matching	Parameters Regression	Loss Function
RegNet [35], 2017	End-to end	PC to Depth ^(1)^	RGB and Depth	Pre-trained ResNet-18, ResNet-18	NiN blocks	MLP ^(4)^	CPL with ED ^(5)^
RGBDTCalib-Net [48], 2022	End-to end	PC to Depth	RGB and Depth	Pre-trained ResNet-34, ResNet-34	CNN blocks	CNN + MLP	CPL with ED
CalibNet [44], 2018	End-to end	PC to Depth	RGB and Depth	Pre-trained ResNet-18, ResNet-18	CNN blocks	CNN + MLP	CPL with ED + photometric loss + PC reprojection loss
CalibRRNet [102], 2023	End-to end	PC to Depth	RGB and Depth	Pre-trained ResNet-18, ResNet-18	CNN blocks	RNN + MLP	Photometric loss + PC reprojection loss
RGGNet [45], 2020	End-to end	PC to Depth	RGB and Depth	Pre-trained ResNet-50, ResNet-50	Feature concatenating layer	MLP	CPL with RD + PC reprojection loss
LCCNet [46], 2021	End-to end	PC to Depth	RGB and Depth	Pre-trained ResNet-18, ResNet-18	Multi-resolution correlation blocks [61]	MLP	CPL with ED + PC reprojection loss
CalibDNN [43], 2021	End-to end	PC to Depth	RGB and Depth	Pre-trained ResNet-18	ResNet blocks	CNN + MLP	CPL with ED + photometric loss + PC reprojection loss
NetCalib [66,101], 2021	End-to end	RGB to Depth^(2)^ PC to Depth	Depth and Depth	Custom CNN blocks	CNN + SPP	MLP	CPL with ED
Calibration-Net [42], 2022	End-to end	RGB to Depth PC to Depth	Depth and Depth	Custom CNN blocks	Multi-resolution correlation blocks	MLP	CPL with ED
LIA-SC-Net [38], 2022	End-to end	PC to Depth, Intensity ^(3)^	RGB and Depth, Intensity	Pre-trained ResNet-18, ResNet-18 with attention mechanism	Multi-resolution correlation blocks	MLP	CPL with ED + structural consistency loss
W. Zhang et al. [68], 2022	End-to end	PC to Depth	RGB and Depth	Pre-trained ResNet-18, ResNet-18	CNN blocks	MLP	Photometric loss + PC reprojection loss
CalibcaNet [83], 2023	End-to end	PC to Depth	RGB and Depth	ResNet-50	Multi-resolution correlation blocks	MLP	CPL with ED
Z. Zhang et al. [70], 2023	End-to end	PC to Depth	LCCNet’s backbone [46]				CPL with ED + PC reprojection loss + supervised contrastive loss
CALNet [39], 2022	End-to end	PC to Depth	RGB and Depth	Pre-trained ResNet-18, ResNet-18	CNN + HSPP + LTC	MLP	CPL with ED + PC reprojection loss
PSNet [61], 2022	End-to end	PC to Depth	RGB and Depth	HRNet-based [86] multi-resolution feature extraction	Multi-resolution feature aggregation + DenseNet	MLP	CPL with ED + PC reprojection loss
CailbRCNN [57], 2020	End-to end	PC to Depth	RGB and Depth	Pre-trained ResNet-18, ResNet-18	CNN + LSTM	MLP	CPL with ED + synthetic view constraint loss + photometric loss + epipolar geometry constraint loss
Causal calibration [64], 2023	End-to end	PC to Depth	RGB and Depth	Pre-trained ResNet-18, ResNet-18	CNN blocks	MLP	CPL with ED + synthetic view constraint loss + photometric loss
SCNet [94], 2024	End-to end	PC to Depth	RGB and Depth	Pre-trained ResNet-18, ResNet-18	Multi-resolution correlation blocks [61]	MLP	CPL with ED + PC reprojection loss
CalibBD [63], 2022	End-to end	PC to Depth	RGB and Depth	Pre-trained ResNet-18, ResNet-18	Multi-resolution correlation blocks [61] + Bi-LSTM	MLP	CPL with ED + photometric loss + synthetic view constraint loss
UniCal [100], 2023	End-to end	PC to Depth	RGB and Depth	MOBILEVIT [99]		MLP	CPL with ED + PC reprojection loss
CalibDepth [47], 2023	End-to end	RGB to Depth PC to Depth	Depth and Depth	Pre-trained ResNet-18	NiN + DCN [93] + LSTM	MLP	CPL with ED + photometric loss + PC reprojection loss
CalibFormer [88], 2023	End-to end	RGB to Depth PC to Depth	Depth and Depth	Multi-resolution feature extraction	Transformer-based feature matching	MLP	CPL with ED + PC reprojection loss
PseudoCal [69], 2023	Hybrid	RGB to PC	RGB and PC	PointPillars [107]	MOBILEVIT [99]	MLP	CPL with ED + PC reprojection loss
CFNet [59], 2021	Hybrid	PC to Depth	RGB and Depth	Pre-trained ResNet-18, ResNet-18	Multi-resolution correlation blocks [61]	RANSAC + EPNP	Photometric loss with calibration flow + structural consistency loss
DXQ-Net [41], 2022	Hybrid	PC to Depth	RGB and Depth	Residual blocks in RAFT [65]	Pyramid 4D correlation-based matching [65]	Gauss Newton method	CPL with RD ^(6)^ + supervised calibration flow loss

(1) PC denotes point cloud, and Depth denotes depth map. (2) RGB denotes RGB images. (3) Intensity denotes intensity map. (4) MLP denotes multi-layer perception. (5) CPL denotes calibration parameter loss, and ED denotes Euclidean distance. (6) RD denotes Riemannian distance.

**Table 2 sensors-24-03878-t002:** Learning-based local feature calibration methods.

Method, Year	Mode	Feature Framework	2D Feature Extraction Block	3D Feature Extraction Block	Feature Objects	Description	Feature Matching/Fusion	Initial Guess	Calibration Model	Optimization
SOIC [36], 2020	Hybrid	Bilateral ^(1)^	WideResNet38 [118]	PointRCNN [121]	Vehicle/Pedestrian/Cyclist (semantic center)	Position/Semantic label		IPPE [134]	Semantic projection consistency loss model	Powell’s conjugate gradient descent
ATOP [40], 2023	Hybrid	Bilateral	ERFNet [122] + MSHA [123] 2D segmentation network (point cloud to depth map)		Person/Other objects (semantic center)	Position/Semantic label/Feature map	Feature map metric	EPnP [103]	Semantic response consistency metric	Particle swarm optimization
SemCal [32], 2021	Hybrid	Bilateral	HRNet + OCR [116]	SalsaNext [117]	Fourteen classes of objects	Position/Semantic label	Semantic-based MI registration	EPnP	Semantic mutual information optimization model	MINE [109]
SemAlign [50], 2021	Hybrid	Bilateral	SDCNet [118]	SPVNAS [119]	Vehicle/Sidewalk/Road	Position/Semantic label		Loss-Guided random sampling	Unidirectional alignment loss model	Adam optimizer
SST-Calib [110], 2022	Hybrid	Bilateral	SDCNet [118]	SqueezeSeV3 [120]	Car	Position/Semantic label			Bidirectional alignment loss model	Adam optimizer
SE-Calib [58], 2023	Hybrid	Bilateral	DFF [124]	3D-SS [125]	Vehicle edge	Position/Semantic label			Semantic response consistency metric model	Grid search [31]
J. Zhu et al. [78], 2023	Hybrid	Bilateral	EfficientDet [126]	SqueezeSeV2 [127]	Patch (window)	Position/Feature map	Cross Model GNN		Jensen–Shannon divergence-based loss + Point projection Euclidean space loss + CPL with ED	
CRLF [114], 2021	Hybrid	Bilateral	BiSeNet-V2 [128]	RANSAC + traditional 3D line fitting	Straight line	Line position/Line equation	Hough transform-based line matching	P3L [135]	Semantic mask-based point cloud projection loss model	Gradient-descent method with annealing
RGKCNet [106], 2022	End to end	Bilateral	Microsoft’s human key point extraction network [129]	PointNet++ [130]	Key point	Position/Intensity/ Feature map	Correlation generation layer		Point projection Euclidean space loss + Jensen–Shannon divergence-based loss	Deep declaration network-based optimization [133]
Y. Zhu et al. [111], 2020	Hybrid	Unilateral ^(2)^	PSPNet [112]		2D car	Position/Semantic label	Semantic mask		Semantic mask-based point cloud projection loss model	Gradient-descent method with annealing
Calib-Anything [52], 2023	Hybrid	Unilateral	Segment anything [113]		2D multi-class objects	Position/Semantic label/Intensity/Normal vector	Semantic mask	Brute force search + random sampling	Semantic alignment loss model	
DedgeNet [108], 2022	End to end	Unilateral	DedgeNet for line extraction	Global feature extraction (ResNet)	2D straight line	Position/Feature map	Correlation generation layer		CPL with ED + point cloud reprojection loss	MLP-based global regression
FustionNet [115], 2023	End to end	Unilateral	Global feature extraction (NiNs)	PointNet++ [130] for key point	3D key point	Position/Feature map	Multi-resolution correlation blocks [61]		CPL with ED	MLP-based global regression

(1) Bilateral denotes bilateral local feature extraction. (2) Unilateral denotes unilateral local feature extraction.

**Table 3 sensors-24-03878-t003:** Calibration results of different methods.

Method, Year	Mis-Calibrated Range	Translation Absolute Error (cm)				Rotation Absolute Error (°)				Robustness
		Mean	X	Y	Z	Mean	Roll	Pitch	Yaw	
Global Feature-Based AEPE										
RegNet [35], 2017	[−1.5 m, 1.5 m]/[−20°, 20°]	6	7	7	4	0.28	0.24	0.25	0.36	☆
RGBDTCalib-Net [48], 2022	[−0.2 m, 0.2 m]/[−10°, 10°]		6.2	2.6	5.8		0.48	0.92	0.3	-
CalibNet [44], 2018	[−0.2 m, 0.2 m]/[−20°, 20°]	4.34	4.2	1.6	7.22	0.41	0.18	0.9	0.15	☆
CalibRRNet [102], 2023	[−0.2 m, 0.2 m]/[−10°, 10°]	1.1347	1.0590	0.9319	1.4131	0.1754	0.0931	0.2290	0.2043	☆
LCCNet [46], 2021	[−1.5 m, 1.5 m]/[−20°, 20°]	1.588	0.243	0.380	0.459	0.163	0.030	0.019	0.040	☆
CalibDNN [43], 2021	[−0.2 m, 0.2 m]/[−20°, 20°]	5.3	6.2	4.3	5.4	0.42	0.18	0.35	0.11	☆
NetCalib [66,101], 2021	[−0.2 m, 0.2 m]/[−2°, 2°]		1.20	2.77	1.10		0.04	0.16	0.09	-
Calibration-Net [42], 2022	[−1.5 m, 1.5 m]/[−20°, 20°]	2.353	2.82	2.35	1.89	0.378	0.105	0.21	0.19	☆
LIA-SC-Net [38], 2022	[−1.5 m, 1.5 m]/[−20°, 20°]	3.96	-	-	-	0.525	-	-	-	☆
W. Zhang et al. [68], 2022	-	25	31	22	22	1.9	2.8	0.2	2.8	☆☆
CalibcaNet [83], 2023	[−0.2 m, 0.2 m]/[−2°, 2°]	6	-	-	-	0.154	-	-	-	☆
Z. Zhang et al. [70], 2023	[−1.5 m, 1.5 m]/[−20°, 20°]	0.33	0.23	0.45	0.30	0.15	0.14	0.13	0.17	☆
CALNet [39], 2022	[−0.25 m, 0.25 m]/[−10°, 10°]	3.03	3.65	1.63	3.80	0.20	0.10	0.38	0.12	☆
PSNet [61], 2022	[−0.25 m, 0.25 m]/[−10°, 10°]	3.1	3.8	2.8	2.6	0.15	0.06	0.26	0.12	-
CalibRCNN [57], 2020	[−0.25 m, 0.25 m]/[−10°, 10°]	9.3	6.2	4.3	5.4	0.805	0.446	0.64	0.199	☆
Causal calibration [64], 2023	[−0.25 m, 0.25 m]/[−10°, 10°]	1.5	2.01	0.88	1.6	0.12	0.0830	0.2291	0.0494	-
SCNet [94], 2024	[−1.5 m, 1.5 m]/[−20°, 20°]	0.724	0.294	0.507	0.480	0.055	0.032	0.039	0.022	☆
MRCNet [87],2023	[−1.5 m, 1.5 m]/[−20°, 20°]	0.35	0.256	0.259	0.535	0.033	0.016	0.046	0.038	☆
CalibBD [63], 2022	[−0.2 m, 0.2 m]/[−20°, 20°]	0.756	0.325	1.066	0.878	0.08	0.021	0.136	0.084	☆
UniCal [100], 2023	[−1.5 m, 1.5 m]/[−20°, 20°]	1.9				0.13				☆☆
CalibDepth [47], 2023	[−1.5 m, 1.5 m]/[−20°, 20°]	1.17	1.31	1.02	1.17	0.123	0.064	0.226	0.080	☆
CalibFormer [88], 2023	[−0.25 m, 0.25 m]/[−10°, 10°]	1.1877	1.1006	0.9015	1.5611	0.1406	0.0764	0.2588	0.0865	-
PseudoCal [69], 2023	[−1.5 m, 1.5 m]/[−30°, 30°]	1.18				0.05				☆
CFNet [59], 2021	[−0.2 m, 0.2 m]/[−20°, 20°]	0.831	0.463	1.230	0.802	0.086	0.028	0.125	0.105	☆
	[−1.5 m, 1.5 m]/[−20°, 20°]	0.995	1.025	0.919	1.042	0.087	0.059	0.110	0.092	
DXQ-Net [41], 2022	[−0.1 m, 0.1 m]/[−5°, 5°]	1.425	0.754	0.476	1.091	0.084	0.049	0.046	0.032	☆☆
Local feature-based AEPE										
SOIC [36], 2020	[−2 m, 2 m]/[−30°, 30°]		4.1	1.5	13		0.075	0.073	0.177	☆
ATOP [40], 2023	-	1.752	0.976	0.222	4	0.08	0.14	0.09	0.02	☆
SemCal [32], 2021	[−0.15 m, 0.15 m]/[−1.5°, 1.5°]	-	3	6	6		0.14	0.17	0.13	☆☆
SemAlign [50], 2021	-/[−20°, 20°]	-	-	-	-	1.49	-	-	-	☆
SST-Calib [110], 2022	[−0.1 m, 0.1 m]/[−20°, 20°]	18.9	-	-	-	0.60	-	-	-	☆
SE-Calib [58], 2023	[−0.1 m, 0.1 m]/[−7°, 7°]	-	-	-	-	0.130	0.095	0.100	0.129	-
J. Zhu et al. [78], 2023	[−0.21 m, 0.21 m]/[−15°, 15°]	1.6	-	-	-	0.15	-	-	-	-
CRLF [114], 2021	[−1 m, 1 m]/[−6°, 6°]		8.2	4.6	9.7		0.216	0.546	0.492	-
RGKCNet [106], 2022	[−0.2 m, 0.2 m]/[−7.5°, 7.5°]	11.7	5	4	5.9	0.43	0.17	0.15	0.16	☆
Calib-Anything [52], 2023	-	10.7	-	-	-	0.174	-	-	-	☆☆
DedgeNet [108], 2022	[−1.5 m, 1.5 m]/[−20°, 20°]	1.109	0.247	0.330	0.153	0.159	0.052	0.014	0.018	☆
REPP										
CalibRank [37], 2020	[−0.28 m, 0.28 m]/[−10°, 10°]		7.01	2.07	6.81		0.17	0.68	0.09	-

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
