# Peer review of "A Review of Deep Learning-Based LiDAR and Camera Extrinsic Calibration"

_sensors, 2024, doi:10.3390/s24123878_

Round 1
Reviewer 1 Report
Comments and Suggestions for Authors
This review article expounds the external parameter calibration of LiDAR and camera based on deep learning, and introduces in the basic framework of external parameter calibration based on deep learning in terms of Accurate calibrating estimation (AEPE) and Relative Calibration Prediction (REPP). The methods used in each step of the basic framework of the two methods are described in detail. The structure of the article is complete and the logic is smooth, but there are still some problems.
This review article expounds the basic principles of LiDAR and camera extrinsic calibration, with a focus on introducing the external parameter calibration of LiDAR and camera based on deep learning, and introduces in the basic framework of external parameter calibration based on deep learning in terms of Accurate calibrating estimation (AEPE) and Relative Calibration Prediction (REPP). The methods used in each step of the basic framework of the two methods are described in detail. The structure of the article is complete and the logic is smooth, but there are still some problems.
Ÿ Line 56-69: please use the correct numbering format.
Ÿ The headings 1.1 and 1.2 were repeated.
The headings of section 1.1 and 1.2 were repeated. The heading of section 1.2 may be incorrect.Ÿ Line 231-246: summary should be done in chronological order?
Ÿ Figure 5: Pipeline of global feature based Calibration?
Figure 5: “Pipeline of global based Calibration” should be written as “Pipeline of global feature based Calibration”.
Ÿ Figure 10: it is not intuitive.
Ÿ References: it is lack of necessary information such as [1], [2], [7], [11], [16], [19], [20], [21], [26], [30], etc.
Reviewer 2 Report
Comments and Suggestions for Authors
1. Figures 1, 2, 7, and 12 should be centered in the paper.
2. No images are shown in Figure 6.
3. There is no image in Figure 8(c).
4. There is no image in Figure 9(a).
5. There is no image in Figure 10.
6. The input and output parts are missing in Figure 15.
7. The output part is not demonstrated in Figure 19.
8. There is relatively little research on REPP methods, and further exploration is needed.
9. Some algorithms need improved robustness in complex environments.
Comments on the Quality of English LanguageMinor editing of English language required
Round 2
Reviewer 1 Report
Comments and Suggestions for Authors
I accept the current version.